# Lung cancer organoids analyzed on microwell arrays predict drug responses of patients within a week

Yawei Hu[1,8], Xizhao Sui[2,8], Fan Song[3], Yaqian Li[3], Kaiyi Li[1], Zhongyao Chen[1], Fan Yang[2], Xiuyuan Chen[2], Yaohua Zhang[3], Xianning Wang[4], Qiang Liu[5], Cong Li[6], Binbin Zou[6], Xiaofang Chen [3,7✉], Jun Wang [2✉] & Peng Liu [1✉]

While the potential of patient-derived organoids (PDOs) to predict patients' responses to anti-cancer treatments has been well recognized, the lengthy time and the low efficiency in establishing PDOs hamper the implementation of PDO-based drug sensitivity tests in clinics. We first adapt a mechanical sample processing method to generate lung cancer organoids (LCOs) from surgically resected and biopsy tumor tissues. The LCOs recapitulate the histological and genetic features of the parental tumors and have the potential to expand indefinitely. By employing an integrated superhydrophobic microwell array chip (InSMAR-chip), we demonstrate hundreds of LCOs, a number that can be generated from most of the samples at passage 0, are sufficient to produce clinically meaningful drug responses within a week. The results prove our one-week drug tests are in good agreement with patient-derived xenografts, genetic mutations of tumors, and clinical outcomes. The LCO model coupled with the microwell device provides a technically feasible means for predicting patient-specific drug responses in clinical settings.

[1] Department of Biomedical Engineering, School of Medicine, Tsinghua University, Beijing, China. [2] Department of Thoracic Surgery, People's Hospital, Peking University, Beijing, China. [3] Key Laboratory for Biomechanics and Mechanobiology of Ministry of Education, School of Biological Science and Medical Engineering, Beihang University, Beijing, China. [4] Beijing OrganoBio Corporation, Beijing, China. [5] Department of Thoracic Surgery, Beijing Haidian Hospital, Beijing, China. [6] Beijing NeoAntigen Biotechnology Co. Ltd, Beijing, China. [7] Interdisplinary Institute of Cancer Diagnosis and Treatment, Beijing Advanced Innovation Centre for Biomedical Engineering, Beihang University, Beijing, China. [8] These authors contributed equally: Yawei Hu, Xizhao Sui. ✉email: xfchen@buaa.edu.cn; wangjun@pkuph.edu.cn; pliu@tsinghua.edu.cn

Despite the increasing availability of therapeutic drugs, lung cancer remains the leading cause of cancer mortality worldwide[1]. This grave situation is, in part, due to the lack of accurate predictions of treatment outcomes for selecting appropriate regimens for patients promptly. Although DNA sequencing has inaugurated an era of precision medicine by linking genetic alterations to targeted drugs, the tumor heterogeneity may confound this gene–drug association[2]. Also, the genotype-based analysis usually falls short in forecasting patients' responses to chemotherapy[3]. As a result, the in vitro cancer model is believed to play an essential role in filling the gap between functional genomics and pathological outcomes. Previously, patient-derived xenografts (PDXs) have been exploited to determine patients' drug responsiveness, but have had limited success due to the low success rates, long turnaround times, and high costs[4].

Recently, patient-derived tumor organoids (PDOs) have emerged as a robust and reliable in vitro tumor model for precision medicine[5–7]. More and more evidence has proved the phenotypic and genotypic concordance between the original tumor tissues and the generated tumor organoids in colorectal[8], pancreatic[9], prostate[10], liver[11], breast[12], bladder[13], and lung[14,15] cancers. Several observational clinical studies have demonstrated that PDOs could deliver a high success rate of prognosing clinical responses of individual patients to therapies in colorectal and gastroesophageal cancer[16–18], leading to the speculation that PDOs may predict treatment response for other types of cancers as well. However, several issues hamper the clinical implementation of PDO-based drug sensitivity tests. First, the current PDO-based drug test still needs quite a few weeks or even months to provide results to patients. This is mainly due to limited numbers of viable organoids derived from patient samples and the use of conventional cell culture techniques, which are operated in the microliter-scale volumes and thus require prolonged in vitro expansion to generate enough quantities of PDOs[19]. Second, the current success rate of establishing organoid cultures with steady expansion rates was still low or the PDO culture conditions have not been established for some types of tumors. For example, PDOs can only be generated from 63% of colorectal cancer (CRC) patients and over one-third of patients could not be benefitted from the PDO-based tests[18]. Despite the quality of the tumor samples, the diversities in niche factor requirements among organoid lines established from different patients may contribute to the low success rate[20,21]. A recent study reported that 87% of lung tumor samples could derive tumor organoids[14], although the numbers and the expansion capacities of these lung cancer organoids (LCOs) were not quantified and whether the lung cancer-derived organoids can prognose clinical response has not been fully explored yet.

To overcome the above-mentioned technical challenges, we envision that one of the most likely ways is to reduce the reaction volumes of the PDO-based drug tests by employing microfabricated array devices that are usually operated on the nanoliter scale. In the current study, we first developed a set of methodologies to derive large numbers of LCOs from patients' samples. We verified that the morphologies, histopathology, DNA CNVs, mutation profiles, and gene expressions of the tumor organoids were consistent with the original tumors and retained even after extended in vitro propagation. We then developed an integrated superhydrophobic microwell array chip (InSMAR-chip) for high-throughput three-dimensional (3D) culture and analysis of LCOs[22,23]. Due to the nanoliter scale of the microwells, organoids obtained at P0 were enough for testing an array of clinically recommended drugs in a week without prolonged expansions. We reported that the responses of LCOs to anticancer drugs were consistent with the clinical outcomes and

genetic mutations. LCOs coupled with the InSMAR-chip may provide an efficient means for predicting patient-specific drug responses in lung cancer promptly.

## Results

**Generation of LCOs from patient samples.** We first established a mechanical sample processing method to generate sufficient numbers of LCOs from patients' tumor tissues in ~3 days. Surgically resected tumor tissues with sizes of ~0.5 × 0.5 × 0.5 cm³ were minced with scissors into small pieces followed by gentle grinding and pushing through a 100-μm strainer with a syringe plunger. Next, the tumor pieces between 40 and 100 μm were collected using a 40-μm strainer and then suspended in an optimized LCO culture medium (LCOM) for overnight culture. After that, the LCOs were seeded in Matrigel and cultured for another 3 days either in multiwell plates for long-term expansions as well as organoid characterizations or on an InSMAR-chip for a 3-day drug sensitivity test (Fig. 1a). During the overnight suspension, the irregular clusters of viable epithelial cells underwent a self-assembly process, transforming to more rounded shapes of tumor organoids with smooth surfaces (Supplementary Fig. 1a and Supplementary Video 1), while the necrotic and fibrosis debris did not show any signs of morphogenesis. We compared the performance of the mechanical processing method with the conventional enzyme digestions by quantifying the number of organoids. All of the five pairs of tumor and normal tissues proved that the mechanical processing generated much higher numbers of organoids from the tumor tissues than the enzymatic digestions (Fig. 1b), probably due to the preservation of the cell–cell connections within cell clusters[24]. When the normal tissues were processed with the mechanical method, the normal lung tissue-derived spheroids (NLSs) were much less than the LCOs obtained from their corresponding tumors (Fig. 1b). By contrast, the enzyme treatment improved the quantities of NLSs, echoing that the epithelial layer was attached to the basement membrane and the stroma in normal tissues, and thus dissolving the extracellular matrix proteins is essential to release the epithelial cells.

We next processed a total of 103 surgically resected lung tumor samples, including 71 adenocarcinomas (ACs), 23 squamous cell carcinomas (SCCs), 4 small cell lung cancers (SCLCs), and 5 other lung cancer types, using the mechanical method. Supplementary Data 1 summarized the LCO numbers, the disease stages, the differentiation status, and other information of all the lung cancer samples. The LCOs showed at least three different morphologies, including solid spheres, luminal spheres, and loosely connected granular sheets (Fig. 1c and Supplementary Fig. 2a), yet there is no obvious relationship between the pathological type and the LCO morphology. The quantities of the LCOs varied dramatically from <10 to >10,000. Again, no significant difference in LCO numbers was observed among lung cancer types or stages (Supplementary Fig. 2b, c). Since the minimum number of organoids required for performing a sensitivity test of one drug on the InSMAR-chip is ~100, we found 55 out of the 71 ACs, 18 out of the 23 SCCs, 4 out of the 4 SCLCs, and 4 out of 5 other lung cancer samples generated >100 organoids, concluding a 79% success rate of sample processing (Fig. 1d). This rate can be further improved by excluding tumor tissues that have serious necrosis, carbon deposition, or fibrosis. Among these 103 specimens, 42 pairs of the tumor and the corresponding normal tissues were processed in parallel. The normal tissues only produced ~15 ± 24 (average ± standard deviation) NLSs, ~200 times less than the numbers of organoids derived from the tumor tissues (3277 ± 4619, average ± standard deviation) (Fig. 1e). We also tried to process eight endobronchial ultrasound-guided transbronchial needle aspiration (EBUS-TBNA) samples

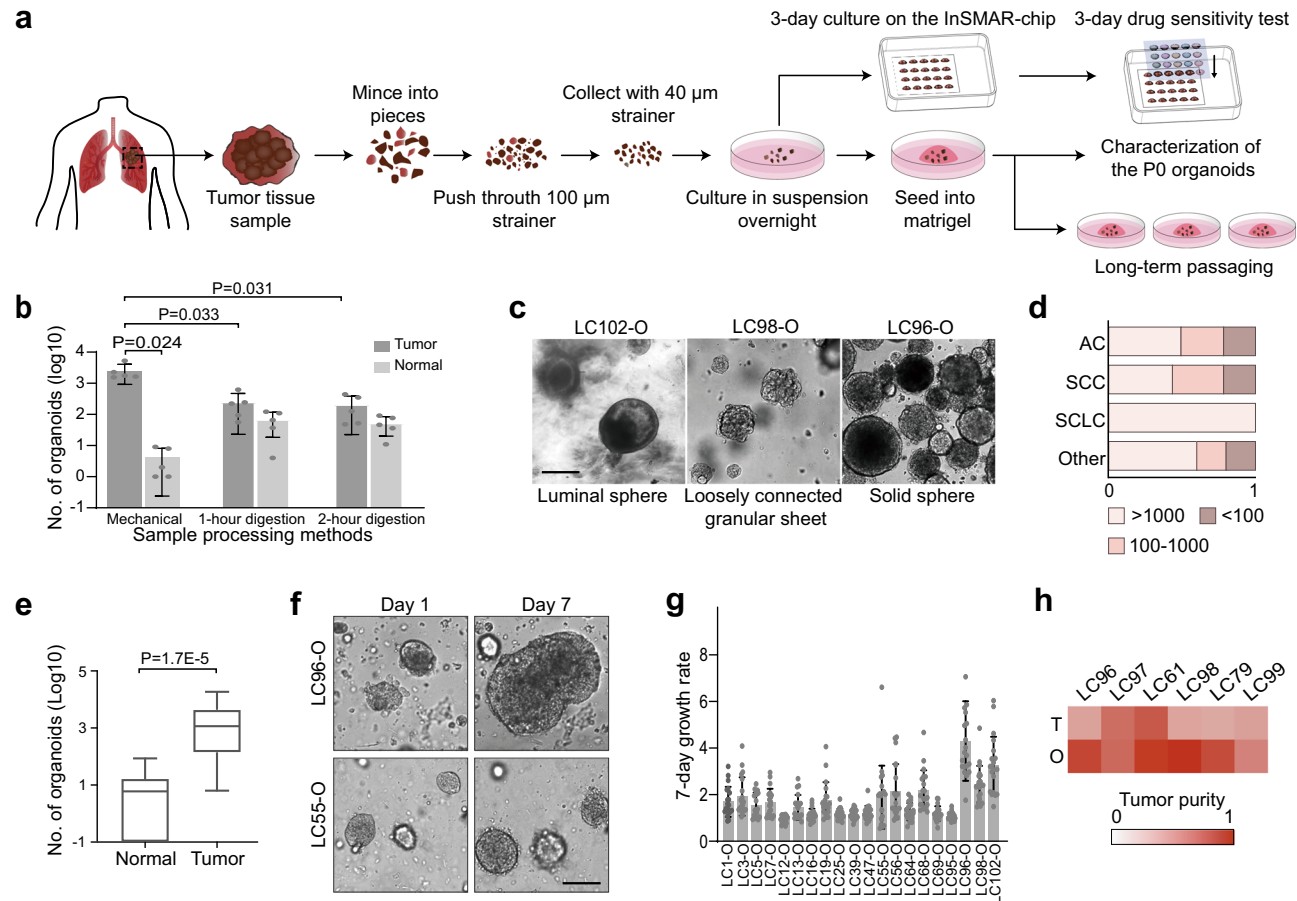

**Fig. 1 Generation of lung cancer organoids (LCOs) from lung tumor tissues. a** Diagram of the process of establishing LCOs from patient tumors for the subsequent long-term culture and the 1-week drug sensitivity test. **b** Bar graph comparing the numbers of organoids generated from tumor and normal tissues using the different methods ($n = 5$ biologically independent samples. Data are presented as mean ± SD. $P$ values are calculated by two-sided Student's $t$ test). **c** Bright-field images of the LCOs with typical luminal sphere (left), solid sphere (right), and loosely connected granular sheet morphologies (middle). The experiments are repeated in 142 patient samples. Scale bars, 200 µm. **d** Stacked bar chart showing the fraction of lung cancer samples that produce <100, 100–1000, and >1000 LCOs by the mechanical processing method. **e** Comparison of the numbers of LCOs generated from tumor tissues and normal lung spheroids (NLSs) generated from paracancer tissues ($n = 42$ biologically independent samples, paired Student's $t$ test. The center line represents the median value. The bounds of box represent the median values of the upper half and the lower half. The bounds of whiskers represent the maxima and the minima. $P$ value is calculated by two-sided Student's $t$ test). **f** Bright-filed images of two LCOs (LC55-O and LC96-O) at days 1 and 7 post seeding in the Matrigel. Scale bars, 200 µm. The experiments are repeated in 142 patient samples. **g** Seven-day growth rates of 20 LCO lines. The 7-day growth rate was calculated by tracking each individual organoid and dividing the area of LCOs at day 7 by that at day 1 ($n = 20$ biologically independent cells, data are presented as mean ± SD). **h** Heat map showing the fraction of cancer cells in the patient tissues and the derived organoids. Note the increased purity of cancer cells in the organoids compared to the original tumor tissues.

using the mechanical method and found that three of them, in which large pieces of tumor tissues with structural integrity were retained, could successfully generate >100 LCOs. (Supplementary Fig. 3).

To further ensure that the tumor organoids were mainly composed of tumor cells, we used a medium not containing the growth factors necessary for the culture of normal lung organoids, such as fibroblast growth factor 7 (FGF7), FGF10, R-spondin, and Noggin[15,25,26]. We closely traced the LCOs generated from 20 samples to quantify the 7-day growth rates in this limited medium. The experiment was performed on the microwell array so that the organoids can be individually observed and the growth rates can be precisely calculated. We found that 19 of these samples showed active growth (7-day growth rate >1), although their growth rates are very diverse (Fig. 1f, g). We also quantified the percentages of cancer cells in six LCO lines based on the next-generation sequencing data, showing that the tumor cell percentages increased from 49 ± 15%

in the tumor tissues to 78 ± 17% in all of the P0 LCOs (Fig. 1h). This evidence demonstrated that our sample processing and culture method can produce cancer-dominant LCOs that are suitable for the subsequent drug sensitivity test.

**Characterization of the LCOs.** The recapitulation of the characteristics of original tumors is the essential feature possessed by the tumor organoids. After the overnight suspension culture followed by the 3-day growth in Matrigel, the LCOs generated by the mechanical processing method soon assumed the morphologies close to those of tumor organoids reported elsewhere[14]. The hematoxylin and eosin (H&E) staining results showed that the LCOs have 3D structures, the same as those of the in vivo tumors (Fig. 2a). LCOs derived from ACs maintained the acinar or solid structures and the expression patterns of thyroid transcription factor-1 and cytokeratin 7 were also retained in the organoids. The SCC organoids LC97-O showed low differentiation, high

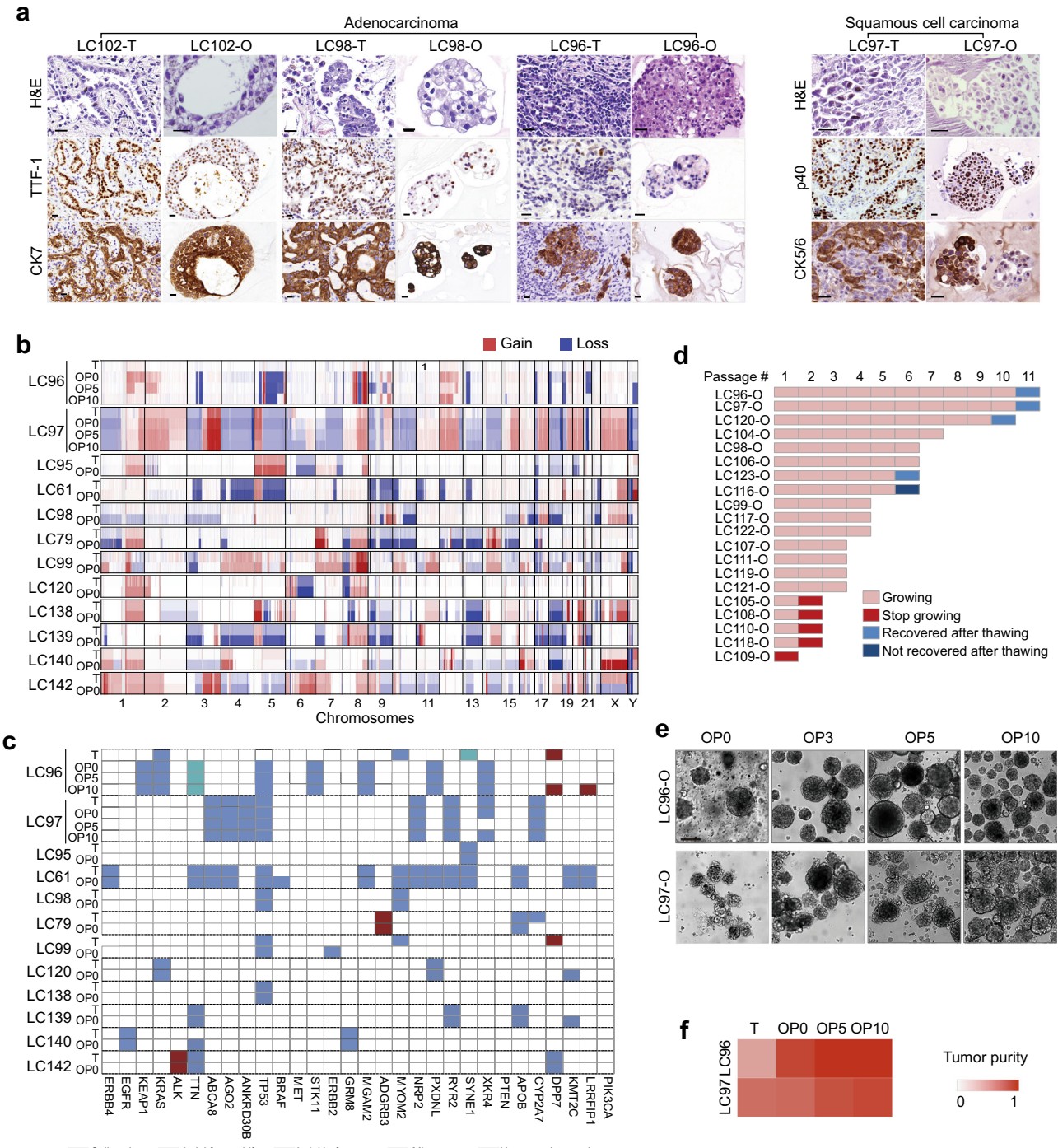

**Fig. 2 Characterization of lung cancer organoids. a** H&E and immunohistochemical staining images of lung cancer tissues and derived organoids. Shown are representative examples of LAC with acinar (LC102 and LC98) or solid (LC96) subtypes and LSC (LC97). The LCOs retained the tumor cell organizations and the expression patterns of the characteristic markers (TTF-1 and CK7 for LAC, p40, and CK5/6 for LSC). Scale bars, 20 μm. The experiments are repeated three times. **b** Heat map illustrating genome-wide copy number variations (CNVs) of lung cancer tissue–LCO pairs. DNA copy number gains (red) and losses (blue) found in the original lung cancer tissues are conserved in the corresponding tumor organoids. Signal amplification can be seen in all the LCOs compared to the original tumor tissues (T tissue, O organoids). **c** Overview of somatic mutations in cancer genes found in the tissue-organoid pairs. Shown is the most severe mutation per gene. **d** Stacked bar graphs illustrating the numbers of passages and the freeze–thaw status of 20 LCO lines underwent long-term culture. Each block indicates one passage or a freeze–thaw cycle. For example, LC96-O and LC97-O have been passaged more than ten times and successfully thawed after frozen at passage 10. LC116-O has been passaged five times but failed to recover after frozen. LC105-O stopped growing at passage 2. **e** Bright-field images showing the unchanged morphologies of LC96-O and LC97-O at P0, P3, P5, and P10. Scale bar, 200 μm. The experiments are repeated three times. **f** Heat map showing the increasing purities of tumor cells with passaging in LC96-O and LC97-O.

proliferation, positively expressed p40, and partial expression of CK5/6, recapitulating the features of the original tumor tissue.

We performed the whole-genome sequencing analysis to characterize the genomes of 12 pairs of lung cancer tissues and their corresponding tumor organoids at passage 0. These samples were selected for sequencing based on the quantities of obtained organoids, which should be enough for performing whole-genome sequencing (WGS) and RNA sequencing (RNA-seq), and the availability of normal tissues. The comparison of the copy number variations (CNVs) in the entire genome as well as major cancer genes, including *TP53*, *PTEN*, *EGFR*, *KRAS*, and *DDR2*, indicates that DNA copy number gains and losses were retained in LCOs, which often showed clearer and more distinct signals than the original lung cancers due to the enrichment of cancer cells (Fig. 2b and Supplementary Fig. 4a)[15]. We also found the mutation loads and the mutation types were mostly conserved in the matching sample pairs, whereas different patient samples showed high diversities in both the total numbers of mutations and the relative contributions of individual signatures (Fig. 2c and Supplementary Fig. 4b, c). In the samples where cancer cell purity increased drastically in the LCOs compared to the original tissue (LC96-O, LC98-O, and LC79-O), more differences can be seen in the mutational profile between the organoids and the parental tissues. We then compared the gene expression profiles of the LCOs with the matching cancer tissues and normal tissues. The correlation heat map shows that the normal tissues are highly correlated with each other, while the organoids cluster with the tumor tissues and the Venn diagrams demonstrate 0.53–0.85 overlap in gene expression between the original cancers and the matching organoids (Supplementary Fig. 5a, c). Kyoto Encyclopedia of Genes and Genomes analysis of the differential genes between LCOs and tumor tissues demonstrated enrichment of genes corresponding to cell adhesion molecules and the immune responses (Supplementary Fig. 5b), consistent with the lack of a tumor microenvironment in organoid culture.

The long-term in vitro expansion capability is one of the most prominent features of tumor organoids, although we may not need the expansion of organoids for drug sensitivity tests in the current study. Here, we cultured the LCOs generated from 16 AC and 4 SCC samples with the mechanical processing method for at least 1 month, discovering the high diversities in the proliferation rates and the expansion capacities (Fig. 2d). Five of the LCOs (3 AC and 2 SCC organoids) demonstrated extremely fast growth rates and were passaged every week (LC96-O, LC97-O, LC116-O, LC120-O, and LC123-O) at a ratio of 1:2. In addition, four of these five LCOs were successfully cryopreserved and thawed and the other one was lost due to the mistakes in cryopreservation. Ten of the LCOs were passaged every 2–4 weeks at least three times, while the other five LCOs stopped growing before passage 3. We also passaged the spheroids derived from normal lung tissues in the LCOM plus FGF7, FGF10, and noggin. These NLSs developed a luminal spheroid morphology and kept growing for >3 months (Supplementary Fig. 6a). During the long-term passaging, the morphologies of the LCOs were preserved and the similar growth curves of LC97-O at P9 and P22 demonstrated that the LCO line retained the expansion capacity (Fig. 2e and Supplementary Fig. 6b, c). The whole-genome sequencing analysis of LC96-O and LC97-O at different passages showed that the CNVs, the driver gene mutations, and the mutation signatures at P0, P5, and P10 were stable (Fig. 2b, c and Supplementary Fig. 4a–c). In addition, the gene expression patterns among different passages of organoids were similar (Supplementary Fig. 5d). During the long-term culture, the percentages of cancer cells in the organoids kept increasing, consistent with the observation that our LCOM only supported the growth of tumor organoids (Fig. 2f). Overall,

the LCOs generated with the mechanical processing method retained the histological and the genetic features of the original tumor tissues and remained stable after prolonged in vitro culture and passage, illustrating that the LCOs at passage 0 indeed possess the major characteristics of tumor organoids.

**Validation of the drug sensitivity test on the InSMAR-chip.** To expedite the process of the drug sensitivity test, we developed an InSMAR-chip to replace the conventional 96-well microplate for culturing LCOs and measuring the responses of LCOs to drugs in the nanoliter scale. Originating from our previous SMAR-chip[22,23], this InSMAR-chip with dimensions of $52 \times 37 \, mm^2$ containing an array of 108 microwells (1.37 mm diameter, 300 μm deep, and 2.25 mm pitch, resulting in a microwell volume of ~440 nl) was made of polycarbonate 2458 by standard injection molding. The 100-μm recessed top surface of the microwell array was filled with a layer of a home-made superhydrophobic paint containing dual-scale titanium dioxide ($TiO_2$) nanoparticles in ethanol-based perfluorooctyltriethoxysilane suspension (Fig. 3a and Supplementary Fig. 7)[27]. Due to the repelling effect of the superhydrophobic surface with a contact angle >160°, a uniform droplet array of culture medium can be spontaneously formed in the microwell array when the excess medium was aspirated out from the chip (Fig. 3b, c and Supplementary Video 2). The operation of this Petri dish-like InSMAR-chip is very flexible. For example, the reagents in the microwells can be changed either as a whole by the submerge-aspirate method or individually by the spot-cover method (Fig. 3d). In addition, since the volume of the microwells is over 400 nl, we found that the Matrigel solution containing a limited number of organoids can be easily loaded into each microwell to form a uniform droplet array using an electronic pipette operated in the multi-dispense mode (Fig. 3c). After gelation of the Matrigel solution, up to 2.4 μl of culture medium can be overlaid onto each gel droplet using the spot-cover method to assist the growth of organoids in this droplet culture mode (Fig. 3c). LCOs cultured on the chip showed similar growth rates and viability as that cultured in the conventional microplate (Fig. 3e–h), and more importantly, maintained the 3D structures of the parental tumor tissue (Fig. 3i). We traced the growth of LCOs on the InSMAR-chip and observed continuous growth for >3 weeks without decline (Supplementary Fig. 8).

Next, we developed a 6-day drug sensitivity test on the InSMAR-chip to evaluate the responses of the LCOs to multiple anticancer drugs (Fig. 4a and Supplementary Fig. 9). To eliminate the variations caused by the uneven numbers and the sizes of the P0 organoids seeded in the microwells, the viability of the LCOs was measured both before (AB-1) and after (AB-2) the drug treatment. The relative cell viability is then represented by the ratio of AB-2 over AB-1 and employed to evaluate the drug responses of LCOs. We compared the alamarBlue™ (AB) cell viability measurements performed on the InSMAR-chip and in the 96-well microplate by culturing and treating A549 cells with different concentrations of doxorubicin. The drug–response curve measured on the chip perfectly overlapped with that in the 96-well plate, demonstrating the reliability of the on-chip cell viability measurement (Fig. 4b, c).

We tested whether the LCOs cultured on the InSMAR-chip responded normally to both the targeted and the chemotherapeutic drugs. The exposure of LCOs to gefitinib (Gef), an epidermal growth factor receptor (EGFR)-tyrosine kinase inhibitor, clearly reduced the activities of extracellular signal-regulated kinase and phosphatidylinositol 3-kinase/AKT downstream of EGFR, inducing apoptosis in LC124-O, which harbors an EGFR P.G719A mutation at exon 18 (Fig. 4d, e). By contrast, Gef did not affect the organoid viability in LC95-O, which has a

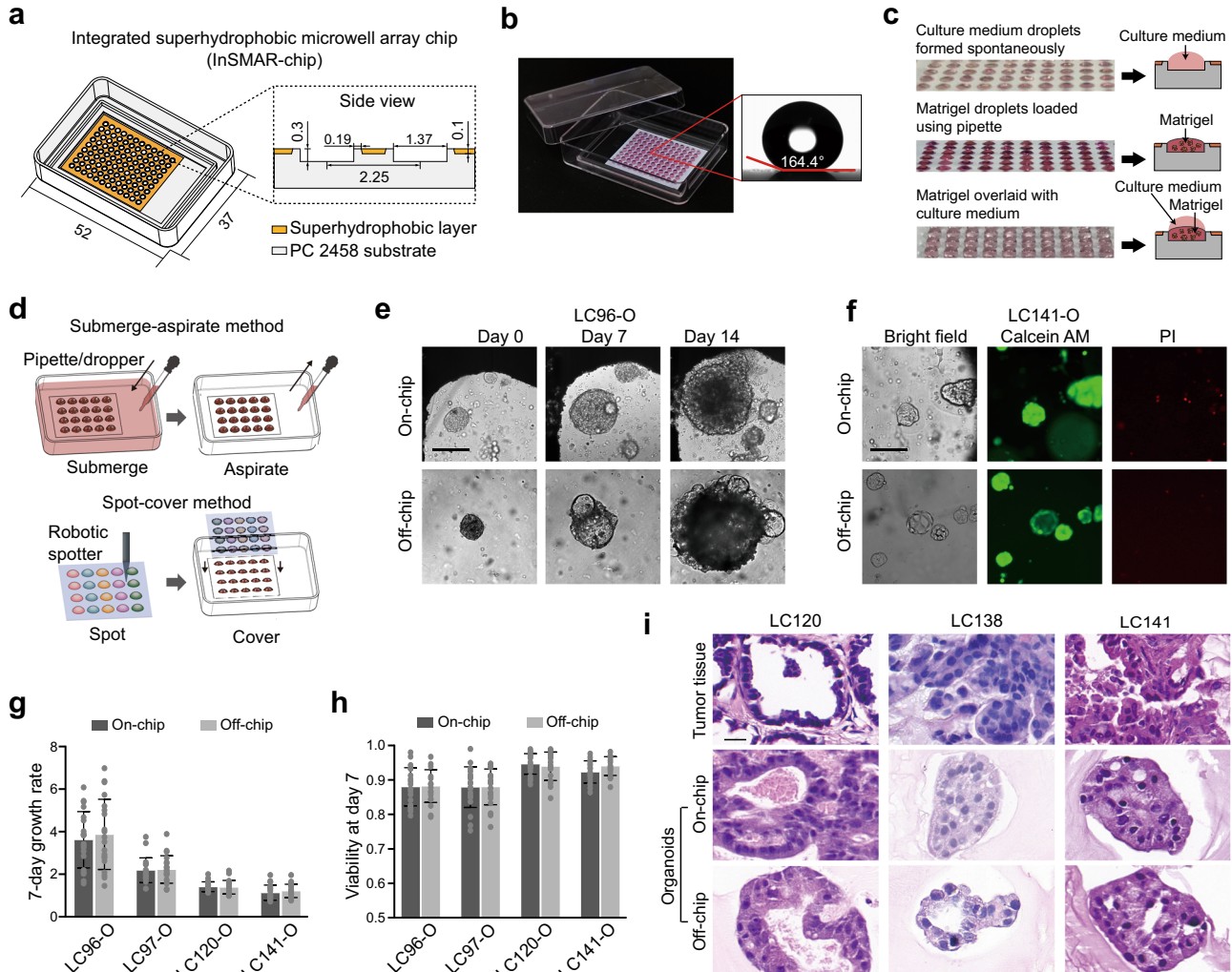

**Fig. 3 Characterization of the integrated superhydrophobic microwell array chip (InSMAR-chip). a** Schematics of the InSMAR-chip (left panel) and the cross-section view of the chip (right panel). **b** Photograph of an InSMAR-chip with a droplet array in the microwells. The contact angle of the superhydrophobic surface is >160°. **c** Photographs of the droplets in the microwells. (Top) The droplet array of culture medium formed spontaneously when the excess medium was aspirated out from the chip. (Middle) The droplet array of the Matrigel loaded into the microwells with an electronic pipette. (Bottom) The Matrigel droplets are overlaid with the culture medium by the spot-cover method. **d** Schematics of the reagent delivery methods on the InSMAR-chip: the submerge-aspirate method and the spot-cover method. **e** Images of LC96-O cultured on the InSMAR-chip (on-chip) and in the conventional microplate (off-chip), showing the continuous growths of LCOs in both conditions from days 0 to 14. Scale bar, 200 μm. The experiments are repeated three times. **f** Bright-field images of LC141-O at day 7 and fluorescent images showing the viability of organoids (green: live cells; red: dead cells). Scale bar, 200 μm. The experiments are repeated three times. **g, h** Comparison of the growth rates (**g**) and the viabilities (**h**) in four organoid lines indicating no significant difference between the LCOs cultured on-chip and off-chip ($n = 20$ biologically independent cells. Data are presented as mean ± SD). **i** H&E stain of the parental tumor tissue and the corresponding LCOs cultured on the InSMAR-chip and in the conventional multiwell plate. Scale bar: 20 μm. The experiments are repeated three times.

wild-type EGFR. Gemcitabine (Gem), a cytidine analog, kills the proliferating cells by inhibiting DNA synthesis and blocking the transition from the G1 to S phase[28]. We performed fluorescence-activated cell sorting (FACS) analysis on LC97-O cultured and treated with Gem on the InSMAR-chip. As expected, the percentage of cells in the S phase decreased from 13.9% before the treatment to 0% after the 24-h exposure (Fig. 4f). In addition to the interruption of the cell cycle, cells treated with Gem also showed the inhibited expression of the antiapoptosis gene, *Bcl2*, and the improved expression of the autophagy-related genes, *Beclin-1* and *LC-3* (Fig. 4g)[29].

Lastly, we validated the results of the LCO-based drug sensitivity test performed on the InSMAR-chip using PDXs, the gold standard of the patient tumor in vitro model (Fig. 4h). In all

three comparisons, the on-chip drug responses of the PDXOs were consistent with the in vivo PDX results (Fig. 4i). PDX1, harboring the Del19 EGFR mutation, was treated with afatinib (Afa), resulting in an increased tumor growth inhibition (TGI) up to 131.52% in 3 weeks. Correspondingly, the viability of the PDX1-derived organoids was reduced to 50% of the dimethyl sulfoxide (DMSO) control under the treatment of 0.21 μM Afa (Fig. 4j). In agreement with the resistance of PDX2 to the chemotherapeutic pemetrexed + cisplatin (PC) and Gem + cisplatin (GC) treatments, PDX2-derived organoids maintained high viability under the exposures to both GC and PC (Supplementary Fig. 10a). The PDX3 groups showed that GC was more effective in inhibiting the tumor growth of the in vivo tumor as well as the derived organoids (Supplementary Fig. 10b).

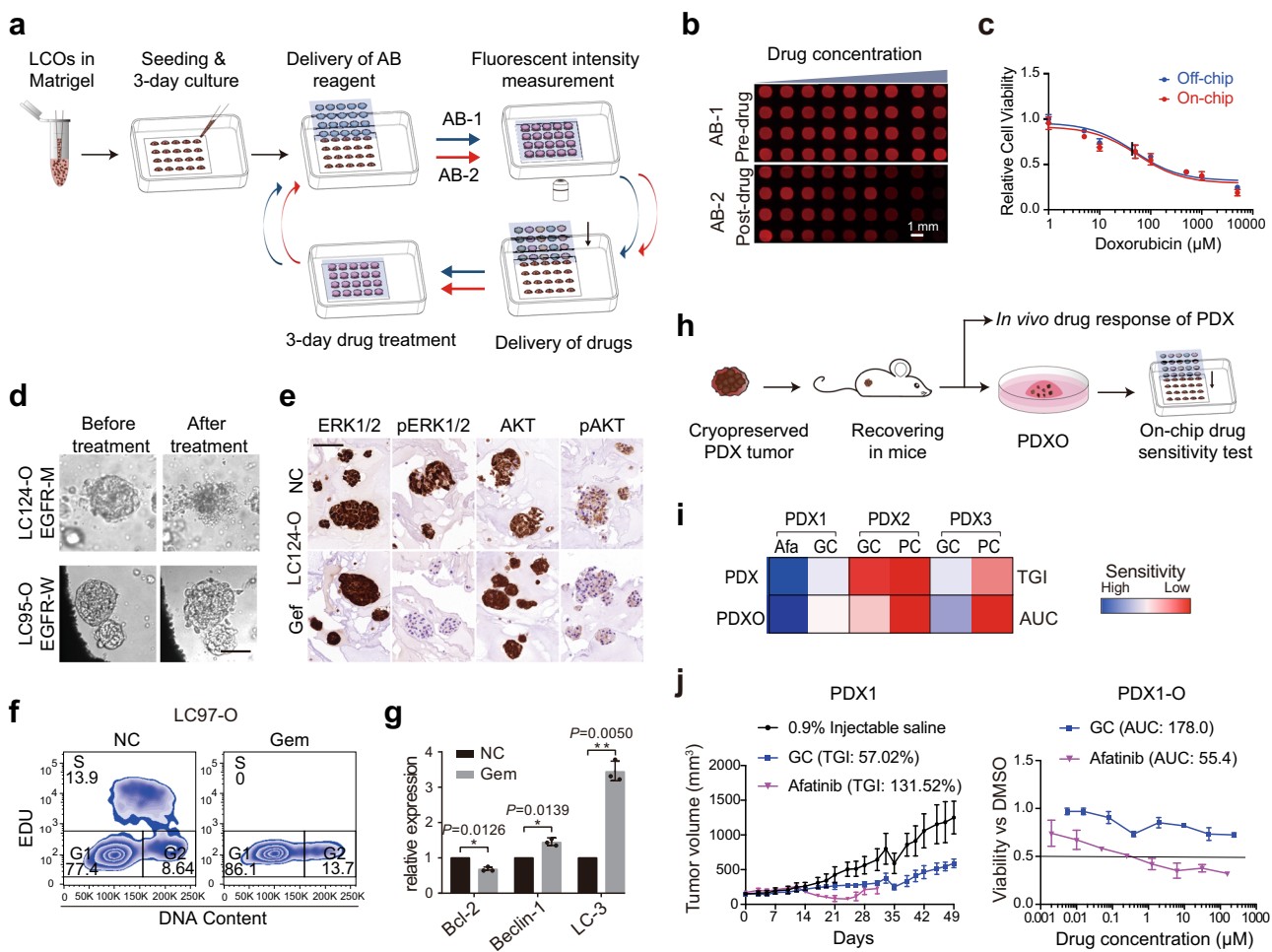

**Fig. 4 Validation of the organoid-based, 1-week drug sensitivity test on the InSMAR-chip. a** Diagram illustrating the procedure of the one-week drug sensitivity test performed on the InSMAR-chip. AB-1: cell viability test with alamaBlue[TM] before drug treatment; AB-2: cell viability test after drug treatment. **b** Mosaic of the fluorescent images of the microwell array showing the fluorescent signals of the microwells before (upper) and after (lower) the drug treatment. Scale bar, 1 mm. **c** Overlapped fitted dose–response curves measured on the InSMAR-chip (on-chip) and in the microplate (off-chip) ($n = 4$ biologically independent cells, data are presented as mean ± SD). **d** Bright-field images of the LCOs treated with gefitinib, demonstrating the organoids with the EGFR P.G719A mutation were killed while the organoids with the wild-type EGFR kept growing under the same condition. Scale bar, 100 μm. The experiments are repeated three times. **e** Images of the immunohistochemical staining indicated that gefitinib inhibits the ERK 1/2 and the AKT activities downstream of EGFR. NC, untreated, Gef gefitinib treated. Scale bar, 200 μm. The experiments are repeated three times. **f** Cell cycle analysis of LCOs cultured on the InSMAR-chip and treated with gemcitabine, showing the elimination of the cells in the S phase by the drug. **g** qPCR analysis of LCOs illustrating the variations in gene expressions upon the gemcitabine treatment ($n = 3$ independent experiments, two-sided Student's $t$ test, data are presented as mean ± SD). **h** Diagram of the comparison process of the in vitro drug sensitivity test using the PDX-derived organoids (PDXO) with the PDX-based drug test in mice. **i** Heat map of the drug effects demonstrating the consistency between the TGI (the tumor growth inhibition) of PDX and the AUC (the area under the dose–response curve) of PDXO. **j** A representative example (PDX1) illustrating that the on-chip drug sensitivity results of PDXO were in good agreement with the responses of PDX mice ($n = 3$ biologically independent animals in the left panel; $n = 3$ biologically independent LCOs in the right panel, data are presented as mean ± SD).

**One-week drug sensitivity tests represent the responses of lung cancers to targeted therapy drugs**. To explore whether the LCO-based 1-week drug sensitivity tests can predict the patient responses to anticancer therapies, we examined the effects of commonly used anti-lung cancer drugs on organoids derived from 21 patient samples (Supplementary Fig. 11a). All the samples were processed using the mechanical method to generate organoids and the subsequent drug assays were performed on the InSMAR-chip within a week from the surgical operation. We first demonstrated that the responses of the LCOs to the targeted therapy were correlated to the genetic mutations of the original tumors using 12 samples[30], in which the information on genetic alterations including EGFR and ALK mutations can be obtained from the clinical data (Supplementary Table 1). Nine LCO lines

are generated from treatment-naive patients, six of which harbor EGFR activating mutations (EGFR-M) sensitive to tyrosine kinase inhibitors and the other three have the wild-type EGFR (EGFR-W), were cultured and exposed to gefitinib on the InSMAR-chips. The drug–response curves of these nine specimens were divided into two groups based on the sensitivities to Gef, perfectly accordant to their genetic mutations (Fig. 5a). We quantified the responses by calculating the area under the dose–response curve (AUC) and the relative viability at the reported plasma trough concentration ($C_{trough}$)[31], both of which were significantly different between the EGFR-M and the EGFR-W groups ($P < 0.0003$ and 0.0002, respectively, Student's $t$ test). The interpolation of the drug–response curves gave half-maximal inhibitory concentration ($IC_{50}$) values between 0.24 and 0.65 μM in the EGFR-M group,

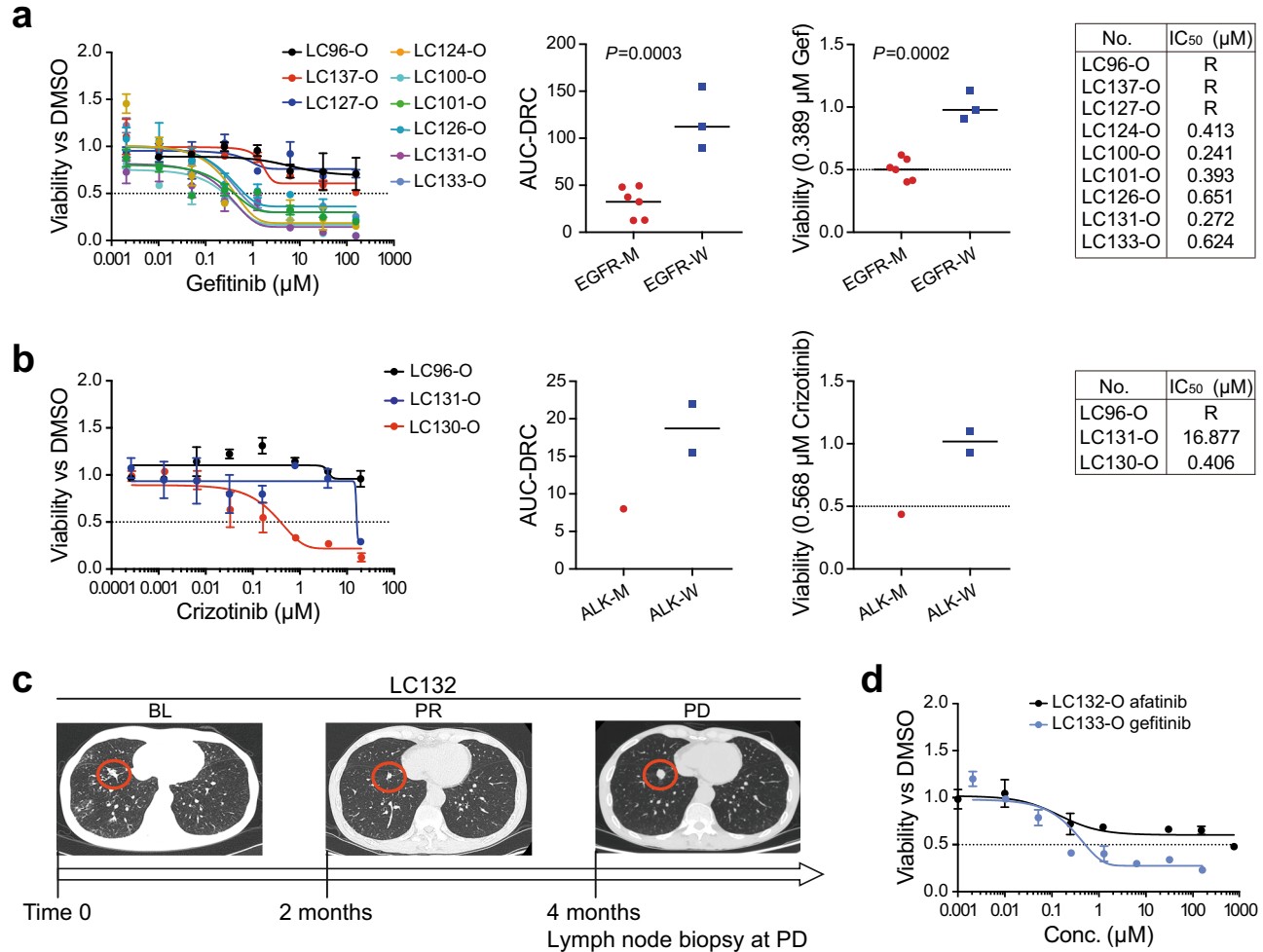

**Fig. 5 LCO-based 1-week drug sensitivity tests reflect the response of lung cancers to targeted drugs. a** Responses of LCOs to the TKI inhibitor, gefitinib, in agreement with EGFR mutations. The fitted dose–response curves (DRCs) represent the viabilities of nine LCOs exposed to a concentration gradient of gefitinib ($n = 3$ biologically independent cells, data are presented as mean ± SD). Three of the LCOs have wild-type EGFR and the other six harbor EGFR activation mutations sensitive to TKI inhibition. For both the area under the dose–response curve (AUC-DRC) and the viability $C_{trough}$, the EGFR mutation (EGFR-M) and the EGFR wild-type (EGFR-W) groups were compared using the unpaired two-tail Student's $t$ test. The IC50 values listed in the table were interpolated from the fitted dose–response curves. R means IC50 is not available since the viability of the LCO is >50% under all the concentrations. **b** Responses of LCOs to crizotinib (ALK inhibitor). The organoid with the EML4-ALK rearrangement mutation (ALK-M: LC130-O) shows reduced viabilities, while those with the wild-type type of ALK (ALK-W: LC96-O and LC131-O) have no responses ($n = 3$ biologically independent cells, data are presented as mean ± SD). **c** CT scan images showing the lung tumor that developed resistance to afatinib as the primary tumor grew (red circles) and cervical lymph node metastases were developed in the course of the treatment. LCOs were generated from the biopsy of cervical lymph node resected at 4 months post the TKI treatment. **d** Fitted dose–response curves illustrating the distinct responses of LC132-O and LC133-O to TKIs, consistent with the patients' responses ($n = 3$ biologically independent LCOs, data are presented as mean ± SD).

falling within the range of clinically relevant concentrations of Gef[32]. We also compared the responses to another targeted drug, crizotinib (Cri), between LC130-O, which harbors the EML4-ALK rearrangement mutation, and the other two ALK-negative LCOs (LC96-O and LC131-O). As we expected, LC130-O was more sensitive to Cri than the other two LCOs, indicated by the lower drug sensitivity curve and the >40 times smaller IC50 value (Fig. 5b).

Next, we verified whether the drug sensitivity tests can recapitulate the patients' responses to TKI therapy. LC132-O was generated from an AC patient who carried the EGFR L858R mutation but acquired resistance to TKI later on. In the first 2 months of the TKI treatment, shrinkage of the primary tumor was evident. However, TKI resistance was developed after 4 months of the treatment with Afa, showing that the primary tumor grew again and the distal cervical lymph node

metastases were detected. The LCOs were generated from the biopsy of the progression disease (PD) lymph node metastasis where the EGFR L858R mutation was still present and the EGFR T790M mutation was absent (Fig. 5c). Another clinical sample, LC133-O, was established from the thoracoscopic biopsy of a treatment-naive patient harboring the EGFR Del19 mutation with N2 lymph nodes metastases and pleural effusion. The first-line treatment of ecotinib (an analog of Gef) resulted in the shrinkage of mediastinal lymph nodes and a partial response (PR) for 4 months. The on-chip drug sensitivity tests of LC132-O and LC133-O produced two widely apart dose–response curves where LC132-O showed more resistance, in great agreement with the clinic outcomes (Fig. 5d). These results suggest that the 1-week on-chip assay reflects the acquired drug resistance of the tumor better than that suggested by the genetic mutations.

**One-week drug sensitivity tests recapitulate patient responses to the chemotherapies**. To explore whether the 1-week drug sensitivity test on the InSMAR-chip can recapitulate the heterogeneous responses of lung tumors to chemotherapies, LCOs cultured on the InSMAR-chip were exposed to three cisplatin-based combinational reagents, PC, GC, and docetaxel + cisplatin (DC), at a concentration ratio of 1:1 ranging from $0.0016 \times C_{max}$ to $125 \times C_{max}$. A total of 14 LCO lines established from AC tissues and 3 LCO lines from SCC samples were tested within a week from the receiving of the samples. Both the AC and the SCC organoids showed striking diversities in their responses to the chemotherapeutic drugs, leading to >1000 times differences in $IC_{50}$ values (Fig. 6a and Supplementary Fig. 11b). On the other hand, the sensitivities of an organoid line to two chemotherapies can be extremely different. For example, in AC organoid lines, LC100-O was more sensitive to GC than PC as indicated by the dose–response curves and the $IC_{50}$ values, whereas LC133-O showed very high sensitivity to PC ($IC_{50} = 0.06 \, \mu M$), but resistance to GC as its viability was not reduced by GC at all (Fig. 6a). Similarly, the responses of the SCC organoid lines, LC134-O and LC125-O, to GC and DC are completely different (Supplementary Fig. 11b). These results emphasized the necessity of the rapid in vitro drug sensitivity test for choosing the most effective chemotherapy for an individual patient.

Next, we compared the chemosensitivity results measured from LCOs on the InSMAR-chip with the corresponding patients' responses in clinics. LC134-O was established from the biopsy of a treatment-naive patient with stage IVA SCC. Four cycles of the dose-reduced chemotherapy with GC resulted in a PR for 7 months (Fig. 6b). The generated LCOs were sensitive to GC, which reduced the viability of the organoids to <50% of the control at the concentration of $0.01 \, \mu M$ (Fig. 6c). LC97-O was established from the resected tumor sample of a patient with stage IIIB SCC. A new metastatic lesion in the ipsilateral station L2 lymph node was discovered right after two cycles of the GC treatment, suggesting the resistance of the original tumor (Fig. 6b). The on-chip sensitivity test of LC97-O also demonstrated the resistance to GC as the organoid viability was not reduced by very high concentrations of GC (Fig. 6c). In addition, we established CRC organoids from the lung metastasis of a CRC patient having PD under oxaliplatin (Oxa) treatment. Again, the on-chip assay represented the patient's response to Oxa as indicated by the slowly decreasing viability under the exposure to the high concentration of Oxa (Supplementary Fig. 11c).

Finally, we advanced the LCO-based on-chip drug sensitivity assay for selecting effective chemotherapeutic regimens on a larger scale as a large number of organoids can be produced by in vitro passaging. We compared the LCO-based screen with the PDX model generated from the same patient sample to demonstrate the reliability of the LCO-based drug selection (Fig. 6d). LC96-O together with the corresponding PDX model was generated from a resected lung AC, which occurred due to rapid bone metastasis and lymph node relapse during the adjuvant chemotherapy with PC (Fig. 6e). The sensitivities to the three cisplatin-based chemotherapies were tested using the expanded LCOs, but none of the drugs can reduce the viability of the organoids effectively. The same chemotherapeutic drugs were administered to the PDX mice and none of them can inhibit the growth of the xenograft tumor efficiently, consistent with the LCO-based screening results (Fig. 6f). Although no effective drugs were successfully identified, these data indicated the feasibility of the LCO-based on-chip drug screen for applications in the precise selection of anticancer drugs for individual patients.

Overall, we performed the drug sensitivity tests in a total of 21 organoid lines on the InSMAR-chips. In ten of the organoid lines (indicated in gray lines in Supplementary Data 2), where the

responses of the respective patients can be evaluated and the administrated drugs were tested by the on-chip assays, the 1-week drug sensitivity results were in great agreement with the clinical data, achieving 100% accuracy and specificity (Fisher's exact test, $P = 0.0048$) (Supplementary Fig. 11d). For the rest of the 11 lines, the relative patient responses were either not comparable to the on-chip test results due to the difference in treatment regimens or not evaluable since adjuvant therapy or no treatment was taken. These results demonstrate the promising potential of the 1-week on-chip drug sensitivity test to predict patient therapeutic responses.

## Discussion

PDOs represent a new generation of in vitro tumor models that can be employed to predict the clinical outcomes of anticancer drugs for individual patients. While encouraging progress has been achieved towards the validation of this new tumor model for precision medicine, the current PDO-based drug sensitivity test still needs several weeks or even months to provide results, mismatching the clinical practices. In the current study, we successfully shortened this process to 1 week by reducing the number of organoids that are needed for the assay and by increasing the number of organoids that can be derived from a patient's sample and proved that the on-chip testing can provide a 100% consistency with the clinical outcomes.

In order to consistently measure drug responses with a high accuracy, the contamination of normal cells in the LCO culture, including both the normal lung epithelial cells and stromal cells such as fibroblasts, should be avoided. In the current study, we collected organoids by filtering the samples through 40- and 100-µm filters (i.e., only the cell clusters in the range 40–100 µm were collected), which can largely decrease the numbers of stromal cells since they generally do not aggregate with other cells. We used a limited medium that inhibits the growths of normal lung organoids and normal epithelial cells. As a result, our organoid cultures were cancer cell dominant, although not 100% pure. In the future, an image-based automatic organoid picking system, which can transfer individual organoids into the microwells directly without getting stromal cells, can be used to further increase the purity of the LCOs in the InSMAR-chip. In addition, a quality control step of morphology checking should be performed to get rid of the microwells or samples with severe stromal cell contamination.

In the targeted therapies, TKIs are generally administered at fixed dosing despite the highly variable outcomes between individuals[33]. As revealed in this work, more than two times variation in the $IC_{50}$ values can be seen in LCOs with identical TKI-sensitive driver gene aberration. These conflicts aroused an interest in us to introduce LCOs into the course of the therapeutic drug monitoring, yet there is still a long way before the relationship between the $IC_{50}$ values of LCOs and the in vitro pharmacokinetics and pharmacodynamics is figured out. Also, our on-chip test could mirror the acquired resistance of a patient to TKI therapy[34], even though the TKI-sensitive mutation L858R was present while the drug resistance indicator T790M was absence[30], highlighting the potential of the organoid-based drug sensitivity test to predict clinical outcomes better than molecular markers.

In the chemotherapies, cisplatin-based adjuvant chemotherapy is routinely used for certain completely resected non-SCLCs[35]. Unfortunately, the conventional response evaluation criteria based on the imaging of tumor sizes are impractical for adjuvant chemotherapy[36] and no molecular markers of lung cancer can be reliably related to the chemotherapy sensitivity[37]. The predictive potential of LCOs would allow the clinicians to prioritize

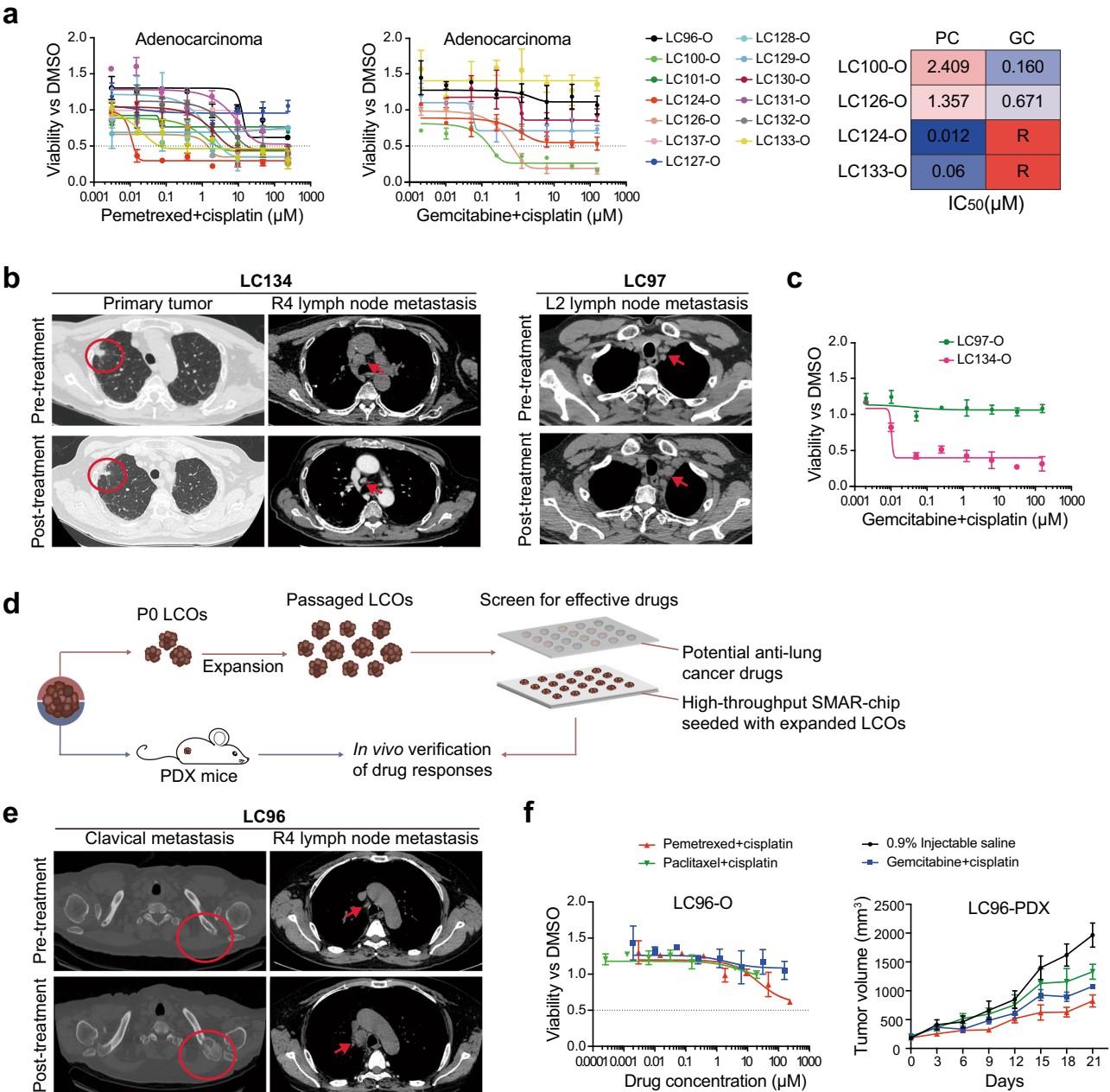

**Fig. 6 One-week drug sensitivity tests represent the response heterogeneity of tumors to chemotherapies. a** Heterogeneous responses of organoids derived from lung adenocarcinomas to the cisplatin-based chemotherapies. The fitted dose–response curves illustrate the responses of the AC organoids to pemetrexed + cisplatin (PC) and gemcitabin + cisplatin (GC) ($n = 3$ biologically independent cells, data are presented as mean ± SD). The heat map on the right are examples of two organoids sensitive to PC and the other two sensitive to GC. R means the $IC_{50}$ is not available. **b** LCO-based on-chip drug sensitivity tests representing the responses of patient tumors to chemotherapies. The CT images on the left are a lung squamous cell carcinoma sensitive to the GC therapy. The red circles indicate the primary tumor and the red arrows point to the metastatic lymph nodes, both of which shrank upon the GC treatment. The CT scan images on the right showed a lung squamous cell carcinoma resists the GC therapy. The red arrows point to the station L2 lymph nodes where new metastasis was discovered after two cycles of adjuvant chemotherapy with GC. **c** Fitted the dose–response curves of the LCOs representing the diverse responses of the in vivo tumors to the chemotherapy ($n = 3$ biologically independent cells, data are presented as mean ± SD). **d** LCO-based on-chip screening of anticancer drugs effective to individual patients. **e** On-chip drug screening using the organoid line (LC96-O). The CT scan images showed the patient tumor metastasized to bone (red circles) and lymph node (red arrows) during chemotherapy with PC. **f** Drug–response curves on the left showing the results of the on-chip screening of three chemotherapies (PC, GC, and paclitaxel + cisplatin) using the LC96-O. The line charts on the right represent the tumor volume of the PDX exposed to the same chemotherapies as the LCOs. Both of the results consistently showed that none of the drug combinations could inhibit the growth of the tumor effectively ($n = 3$ biologically independent LCOs in the left panel; $n = 4$ biologically independent animals in the right panel, data are presented as mean ± SD).

adjuvant chemotherapies prospectively by comparing the responses of the generated LCOs to different drugs, although a more well-designed pilot study should be launched to assess the sensitivity and specificity of the assay.

Considering the EBUS-TBNA is the main technique to obtain tumor specimens for unresectable stage IIIA-N2 or M1 lung cancer[38], we have tried the generation of LCOs from EBUS-TBNA samples and achieved a success rate of ~40%. We found that the quality of the biopsy sample was crucial for the successful establishment of LCOs, as samples with low cellularity or seriously damaged structure were usually associated with the failure of organoid formations. A rapid onsite evaluation of the biopsy adequacy by a pathologist will be helpful to improve the sample processing[39]. In addition, the culture medium innovation to enable the formation of organoids from single cancer cells would also improve the success rate of this type of challenging clinical samples.

Immune checkpoint inhibitors have been recommended as the standard single-agent or combinational therapy for patients without driver gene aberrations[40]. Due to the lack of reliable and dynamic predictive biomarkers for PD-1 inhibitors, such as pembrolizumab, the wide applications of immunotherapy are hindered[41]. While tumor organoids recapitulate the characteristics of the in vivo tumors, the absence of the integrated tumor microenvironment is the main obstacle for employing organoids in the research of immunotherapy. We expect to introduce a coculture system[42] to mirror the tumor microenvironment for the next step.

In conclusion, we established a set of feasible approaches to shorten the LCO-based drug sensitivity test to a week. The tests of patient samples demonstrated that the drug responses reported using our method correlated greatly with genetic mutations and clinic outcomes. Furthermore, the screening of chemotherapy drugs using the expanded LCOs on the InSMAR-chip demonstrated good consistency with PDX models. These results validate the potential of our platform for predicting the responses of in vivo tumors to anticancer reagents and for screening the most effective drugs in personalized cancer treatment.

## Methods

**Study design**. The objective of the study is to determine whether the on-chip LCO-based drug sensitivity test can represent the patients' responses to standard-of-care treatments. The collection of patient tissues and data for the generation of LCOs and the LCO-based drug sensitivity tests were approved by the ethical review boards of the Peking University People's Hospital. Main inclusion criteria included patients with clinically local advanced or metastatic lung cancer, aged 18 years or older, fresh tissues available through either biopsy or surgical resection of the primary or metastatic lesions, and enough numbers of organoids generated from patients' samples. Candidates were assessed to determine eligibility and informed consent was obtained before operation. The drug sensitivity tests were performed on the InSMAR-chips once the organoids were successfully established within a week after the operation. However, only patients with concurrent or subsequent clinical drug–response evaluations, which were comparable with the on-chip assays, were enrolled in the clinical study.

**Processing of lung cancer tissues**. All the lung cancer samples were collected at the Peking University People's Hospital and approved by the local Ethics Committee. Written informed consent was obtained from the patients and/or their authorized representatives. Lung cancer tissues stored in a preservation solution (Dulbecco's modified Eagle's medium/Nutrient Mixture F12 (DMEM/F12) containing 1% penicillin and streptomycin (Gibco)) were transported to the laboratory and processed within 1 day of removal from the patients. Upon arrival, the lung cancer tissues were first photographed, weighed, and volume measured. After that, the tissues were cut into several 1–5 mm$^3$ pieces with no bias. One piece was quickly put into a freeze-storage tube and stored at −80 °C for the subsequent whole-genome DNA sequencing. One piece was preserved in the RNAlater solution (Qiagen) for the subsequent total RNA isolation. One piece was fixed in 4% paraformaldehyde (Sigma-Aldrich) for histopathological and immunohistochemical staining. The remainder was minced as small as possible with surgical scissors and then resuspended in 10-ml Advanced DMEM/F12 containing 1% penicillin and streptomycin. The suspension was filtered through a 100-μm strainer (Falcon)

and the tissue masses on the membrane were gently ground and pushed through the strainer with a 5-ml syringe plunger. After the membrane rinsing, the residual impurities on the membrane were discarded together with the strainer. Next, the filtrate containing cell clusters and single cells was strained through a 40-μm strainer to collect the cell clusters with sizes between 40 and 100 μm. The filter membrane was cut off from the strainer, placed into 2 ml of LCOM, and washed thoroughly with a pipette to release the cell clusters into the media. Then, the membrane was discarded and the cells were cultured in suspension overnight. The procedures of processing paracancerous normal tissues and xenografts were the same as those described above. During the enzymatic digestion, the small pieces of tumor tissue were digested in 1% collagenase for 1 or 2 h, followed by the collection of cell clusters between 40 and 100 μm.

**Culture, passaging, and cryopreservation of lung cancer organoids**. To culture LCOs in a multiwell plate, LCOs in suspension were first centrifuged for 5 min at $500 \times g$ in 4 °C and resuspended in the cold growth factor-reduced Matrigel (BD Biosciences). Then, 60-μl drops of the Matrigel cell cluster suspension were inoculated into an ultra-low attachment 96- or 48-well plate with a flat bottom (Corning) and were allowed to solidify at 37 °C for 20 min. The seeding density was adjusted to ~500 organoids per well. Once the Matrigel became stable, 100 μl of LCOM was added into the wells and the plate was transferred to a cell culture incubator at 37 °C with 5% $CO_2$. The LCOM was refreshed twice per week. The recipe for LCOM is listed in Supplementary Table 2. When organoids were cultured on an InSMAR-chip, 0.6-μl Matrigel containing 5–10 organoids were loaded into each microwell with an electronic pipette (Rainin E4 XLS, Mettler-Toledo) working in a low-speed multi-dispense mode. Since the volume of each microwell was only 440 nl, the Matrigel formed an extruding meniscus outside of the well. In the submerging culture mode, 5-ml LCOM was added into the InSMAR-chip to submerge the entire microwell array and no culture medium exchange was needed during the 3-day culture. In the droplet culture mode, each Matrigel droplet in the microwell was overlaid with 2.4-μl LCOM using the spot-cover method. Due to the superhydrophobic property of the InSMAR-chip, the LCOM formed a rounded drop attached to the Matrigel without mixing between adjacent microwells. The LCOM was refreshed every day by sweeping away the old medium with filter paper and overlaying 2.4-μl fresh medium to each microwell again.

Passaging of LCOs was performed using the mechanical method in the microplate. Briefly, LCOs were harvested and digested in 10× volumes of cold Organoid Harvesting Solution (R&D Systems) on an orbital shaker at 0 °C for 2–3 h to dissolve the Matrigel. Once the Matrigel was digested completely, the organoid suspension was sheared by pipetting, followed by washing with Advanced DMEM/F12, centrifugation (500 × g, 5 min, 4 °C), suspension in the Matrigel, and re-seeding at the ratio of 1:2–1:4. Some passages with sufficient quantities can be bio-banked in the Cryopreservation Medium (CELLBANKER 2, Thermo Fisher).

**Histology and immunostaining**. The harvested LCOs were washed with cold phosphate-buffered saline (PBS), suspended in 40 μl of 10 mg/ml fibrinogen solution (Sigma-Aldrich), and then immediately mixed with 20 μl of thrombin reagent (Solarbio) for fibrin polymerization. After that, the fibrin hydrogel containing organoids together with the matched tissues were fixed in 1 ml of 4% paraformaldehyde (Sigma-Aldrich), followed by dehydration, paraffin embedding, sectioning, and a standard H&E staining protocol. For immunohistochemistry staining, the paraffin slides were first baked at 72 °C for 30 min, followed by deparaffinization in xylene, rehydration through a graded ethanol series, and boiling in an EDTA 9.0 solution. The endogenous peroxidase was blocked in 3% hydrogen peroxide for 15 min. The slides were washed in PBS and incubated with primary antibodies at 4 °C overnight. After a wash with PBS, slides were incubated with secondary antibodies at room temperature for 20 min and developed with a DAB chromogenic solution (OriGene) following the manufacturer's instructions. Cell nuclei were counter-stained with hematoxylin. Then, slides were dehydrated, hyalinized with xylene, and sealed for microscope snapshot. More detailed information on antibodies was listed in Supplementary Table 3. Bright-field and immunofluorescence images were obtained using the Olympus IX83 inverted fluorescence microscope. H&E and immunohistochemistry images were acquired using the 3DHISTECH Panoramic SCAN system.

**DNA-sequencing analysis**. For the whole-genome sequencing analysis, DNA from parental tumor tissues, blood or paracancerous tissues, and matched tumor organoids were isolated using the DNeasy Blood & Tissue Kit (Qiagen) supplemented with the RNase Inhibitor (Invitrogen) treatment following the manufacturer's protocol. The quality and the concentration of DNA were assessed using the Nanodrop (ND-100) (260/280 ratio) and measured using the Qubit dsDNA HS Assay Kit (Invitrogen). For DNA library preparation, briefly, 500 ng DNA of each sample was fragmented using the Ultrasonic DNA shearing (S220, Covaris). The desired length range of the DNA fragments was purified using the AMPure XP beads (Beckman Coulter) and checked with the Agilent 2100-chip (Agilent). The recovered DNA was used to generate DNA libraries using the Ultra II DNA Library Prep Kit (NEB). The paired-end (2 × 150-bp) WGS was conducted on the Nova-Seq6000 (Illumina) by CapitalBio Technology. The sequencing data were processed using our in-house developed somatic mutation analysis workflow according to

best practices guidelines for the Genome Analysis Toolkit (GATK) v.4.1.0.0[43]. The sequencing reads were aligned to the human reference genome hg19 using the Burrows-Wheeler Alignment with maximal exact matches v.0.7.16a[44]. Then, the SAMBLASTER v.0.1.24[45] and the SAM tools v.1.9[46] were used to mark duplication reads. Somatic variants were obtained using the Mutect2[47] in the GATK and the filtration of the mutations was performed using the FilterMutectCalls. The germline variants were obtained using the HaplotypeCaller in the GATK and the high-quality germline variants together with training resources from HapMap, 1000G, Omni, and dbSNP138 provided by the GATK were processed using the Variant Quality Score Recalibration. The purities of the tumor tissues were computed according to the sclust v.1.0[48]. The CNVkit v.0.9.3[49] was used to obtain the germline and the somatic CNV data for comparative analyses. The mutational signatures were analyzed using an R package named the BSgenome[50].

The clinical data of the genetic alterations including EGFR mutation and ALK fusion were identified using a capture-based targeted sequencing panel that consisted of 520 or 168 cancer-related genes (Burning Rock Biotech, Guangzhou, China) and 457 or 31 cancer-related genes (Berry Oncology, Fujian, China) by next-generation sequencing as previously described[51].

**RNA-seq analysis of LCOs.** Total RNA from tumor tissues and blood or para-cancerous tissues was extracted using the RNeasy Mini Kit (Qiagen) and messenger RNA (mRNA) was captured using the Poly (A) mRNA Magnetic Isolation Module (NEB). The mRNA from matched organoids was extracted directly using the Dynabeads mRNA DIRECT Kit (Invitrogen). The quality and the quantity of mRNA were assessed and measured with the Agilent 2100-chip and the Qubit RNA HS Assay Kit (Invitrogen), respectively. The mRNA libraries were generated with 20 ng of initial mRNA for sequencing using the Ultra II RNA Library Prep Kit (NEB). The paired-end ($2 \times 150$-bp) RNA-seq was performed on the NovaSeq6000 by CapitalBio Technology. RNA-seq data were analyzed using the miARma v.1.7.3 pipeline[52]. The sequencing qualities were assessed using the FastQC (v.0.11.5)[53] software. The trimmed reads (trimmed $5',3'$-adaptor bases using the Cutadapt v.2.1[54]) were aligned to the reference genome hg19 using the STAR software (2.5.3a)[55]. The gene abundances for each sample were estimated with the. The RPKM (Reads Per Kilobase of transcript per Million) values of genes and transcript levels were calculated using the edgeR package (v.3.24.3)[56], which was also used to calculate the differentially expressed genes.

**Fabrication of the InSMAR-chip.** To prepare the superhydrophobic paint, 1 g of $1H$, $1H$, $2H$, $2H$-perfluorooctyltriethoxysilane (Sigma-Aldrich) was added into 99 g of absolute ethanol and mechanically stirred for 2 h. Then, 6 g of titanium oxide nanoparticles (~60 to 200 nm) (Sigma-Aldrich) and 6 g of P25 TiO₂ (~21 nm) (Degussa) were added into the solution to make a paint-like suspension, which was sonicated for 30 s to disperse the particles. After that, the suspension was pipetted onto the recessed top surface of the InSMAR-chip, which was manufactured by the standard injection molding. The paint was air-dried completely within 30 s and the InSMAR-chip was autoclaved and sealed in a plastic bag until use.

**Cell proliferation assay.** The growth rate of organoids was measured following the method described previously[57]. Briefly, the organoids were dissociated into single cells and 100,000 cells as initial setting were seeded into a 48-well plate, in triplicate. Cells encapsulated in the Matrigel were cultured in LCOM for 8 days, then newly grown organoids were digested into single cells again, and the number of cells was counted with a hemocytometer and trypan blue exclusion. The growth rate was calculated from the mean of three replicates using the following equation:

$$y(t) = y_0 \times e^{(\text{growth rate} \times t)}$$

where $y(t)$ is the number of cells at the final time point, $y_0$ is the number of cells at the initial time point, and $t$ is the time.

**Quantitative RT–PCR.** Total RNA was extracted from LCOs using the RNeasy Mini Kit (Qiagen) following the manufacturer's instructions. After that, complementary DNA (cDNA) was synthesized in a volume of 10 µl using 0.5 µg of total RNA with the ProtoScript II First Strand cDNA Synthesis Kit (NEB). Then, 2 µl of cDNA was amplified with the PowerUp™ SYBR Green Master Mix (Thermo Fisher) using the gene-specific primers listed in Supplementary Table 4. The reactions were run in the CFX96 Touch Real-Time PCR Detection System (Bio-Rad) using a standard thermal cycling program (35 cycles at 95 °C for 20 s, 58 °C for 20 s, and 72 °C for 45 s) with three replicates for each sample. The relative mRNA levels of the target genes were analyzed using the $\Delta\Delta C_T$ method with an internal reference gene of β-actin.

**Cell cycle analysis.** LCOs were treated with 10 µM of Gem, and vehicle (0.1% DMSO) for 24 h, followed by incubation with 1× EdU medium for 2 h. After that, LCOs were harvested and dissociated into single cells using an Organoid Harvesting Solution (R&D Systems) combined with TrypLE (Life Technologies). Dissociated single cells were fixed in 4% paraformaldehyde for 15 min and permeated with 0.3% Triton X-100 for 12 min at room temperature. After a wash with 3% bovine serum albumin, cells were stained using the BeyoClick™ EdU Cell

Proliferation Kit (Beyotime Biotechnology). The stained cells were washed and suspended in 500 µl of PBS and analyzed using the BD FACS Aria II flow cytometry (BD Biosciences).

**One-week drug sensitivity test performed on the InSMAR-chip.** Following the overnight culture in suspension, the LCOs were harvested, suspended in 1 ml of Advanced DMEM/F12, and counted. After that, the LCO suspension was centrifuged and resuspended in 30 µl of Matrigel to a concentration of 10–15 LCOs/µl, followed by the inoculation of LCOs with the Matrigel on an InSMAR-chip using the electronic pipette. After the solidification of the Matrigel, 2-ml LCOM was dispensed into the InSMAR-chip to submerge the microwell array. The LCOs were cultured in the submerging culture mode for 3 days in the cell incubator at 37 °C with 5% CO₂.

On day 3, the first cell viability assay (AB-1) was conducted using the alamarBlue™ Cell Viability Reagent (AB reagent, Invitrogen). After most of the LCOM was aspirated out from the chip, a piece of filter paper was swept across the microwells to remove excess medium droplets that were adsorbed onto the Matrigel. According to the spot-cover method, the AB reagent was dispensed on a silylated glass slide to form an AB array with a droplet volume of 600 nl at a concentration of 10% using a robotic spotter (PersonalArrayer 16, CapitalBio). The slide with the AB array was then covered onto the microwell array of the InSMAR-chip upside down. The InSMAR-chip overlaid with the AB array slide was transferred to the cell incubator for a 2-h incubation. After that, the InSMAR-chip was scanned and the fluorescence signal was measured using the Olympus IX83 inverted fluorescence microscope. The covered slide flattened the top of the droplets in the microwells and eliminated the rings of light around the droplets during imaging. The fluorescent intensity of each microwell was measured using the ImageJ software. To eliminate the background noise introduced by the AB reagent itself, we incubated the AB reagent with the Matrigel in the microwell without LCOs as a negative control (NC). After scanning, AB reagent was removed using the submerge-aspirate method, followed by sweeping with filter paper. Next, an array of drugs with a droplet volume of 2.4 µl was delivered to the InSMAR-chip using the spot-cover method. Each drug was tested at eight different concentrations with three repeats. As a result, four drugs can be tested together on a single InSMAR-chip containing 108 microwells. The detailed information on the drugs was listed in Supplementary Table 6. All of the 15 Food and Drug Administration-approved therapeutic agents were stored as 10 mM in DMSO at −20 °C. The drug gradient with eight concentrations was prepared in a 5-fold serial dilution. In the combinatorial drug treatment, the chemotherapeutic and the platinum agents were mixed at a concentration ratio of 1:1. The concentrations of the drugs for dispensing on the drug array were calculated as follows: drug dispensation concentration = (final microwell concentration) × 3/2.4.

The LCOs in the microwells were treated with drugs for 72 h in the droplet culture mode before the second cell viability test (AB-2) was conducted. At the end of the treatment, the drugs in the microwells were removed using the submerge-aspirate method and a new AB array was aligned and covered onto the InSMAR-chip again. After the 2-h incubation, the InSMAR-chip was scanned and the values were obtained as described above. The percentage of viable cells after the drug treatment was analyzed by normalizing AB signals with that of the vehicle control (0.1% DMSO, VC) and NC as described in Eq. (1):

$$\text{Cell viability} = \left[\frac{O_{AB-2} - NC_{AB-2}}{O_{AB-1} - NC_{AB-1}}\right] \Bigg/ \left\{\frac{\sum[(VC_{AB-2} - NC_{AB-2})/(VC_{AB-1} - NC_{AB-1})]}{n}\right\} \quad (1)$$

where $O_{AB-1}$ and $O_{AB-2}$ are the fluorescent intensities of drug-treated organoids measured from the first and the second cell viability assays, $NC_{AB-1}$ and $NC_{AB-2}$ are the background noise values of NC from the first and the second cell viability assays, $VC_{AB-1}$ and $VC_{AB-2}$ are the fluorescent intensities of VC from the first and the second cell viability assay, and $n$ is the number of the repeated microwells at each drug concentration. The dose–response curves were plotted as the percentage of the cell viability against the logarithm of drug concentrations in µM and were fitted to estimate the $IC_{50}$.

**Animals and the establishment of tumor xenografts.** Six- to eight-week-old male NOD-Prkd$^{scid}$I12rg$^{em2Idmo}$ (NPI) mice were maintained in an environment with a temperature of 22 ± 1 °C, relative humidity of 50 ± 1%, a light/dark cycle of 12/12 h, and provided with sterile food and water from Beijing IDMO Co., Ltd. All animal studies were performed with approval from the Animal Care and Use Committee of the People's Hospital of Peking University. In the process of the construction of PDX mouse models, fresh tissues were transported to IDMO for xenografting at 4 °C in a tissue conservation medium within 48 h. Upon arrival, the samples were spliced into one to four $3 \times 3 \times 3$ mm³ pieces, washed twice with PBS, and injected into the back of three male NPI mice subcutaneously in 100 µm of PBS/Matrigel (1:1 ratio), marked as the PA generation. Forming the PA tumor tissues, necrosis and unusable parts of the tumor tissues were removed, the remaining tissues were cut into small pieces, and transplanted into the mice, marked as P0 generation. The subsequent passaging operation is similar to that of the P0 tumor tissues. All of the animals were anesthetized with 15 mg/kg of Zoletil® and 2.5 mg/kg of Rompun® by intraperitoneal (i.p.) injection for tumor implantation. Mice with stable P1 tumor tissues (average volume of ~100–200 mm³) were treated with vehicle (0.9% injectable saline, 1 mg/kg, i.p. b.i.w.), Gem (100 mg/kg,

i.p. b.i.w.), Pemetrexed (100 mg/kg, i.p. b.i.w.), or Afa (20 mg/kg, p.o. q.d.) for 3 weeks. Following the implantation, the mice were monitored twice per week. The tumor volumes were calculated using the equation of $(L \times W^2)/2$, where $L$ is the longest and $W$ is the shortest axis of the tumor. TGI was calculated as in Eq. (2):

$$\text{TGI} = \left[1 - \left(T_i - T_0\right)/\left(V_i - V_0\right)\right] \times 100\% \qquad (2)$$

where $T_i$ is the mean tumor volume (TV) of the treatment group on the measurement day, $T_0$ is the mean TV of the treatment group at the first day of dosing, $V_i$ is the mean TV of the control group at the measurement day, and $V_0$ is the TV of the control group at the first day of dosing. Once the xenograft tumor had attained a size of 1000 mm$^3$, the tumor was excised and the mice were euthanized following the protocol of the Laboratory of Animal Research Center of Tsinghua University. The xenografts were subsequently transplanted from mouse to mouse and stocked frozen in a DMSO-fetal calf serum solution or frozen-dried in nitrogen for further studies.

**Generation of PDX-derived organoids**. Cryopreserved PDX tumor samples from previously established lung tumor PDX models were recovered and implanted into NPI mice. When the tumor volume reached 800–1000 mm$^3$, the tumor was harvested and sliced into small pieces and further transplanted to NPI mice for in vivo drug sensitivity tests, which were initiated once the transplanted tumor reached 100–200 mm$^3$. Tumor tissues from the mice of the same generation were harvested to generate PDX-derived organoids using the method described in the previous session.

**Reporting summary**. Further information on research design is available in the Nature Research Reporting Summary linked to this article.

## Data availability

The raw sequence data reported in this paper have been deposited in the Genome Sequence Archive (*Genomics, Proteomics & Bioinformatics*, 2017) in National Genomics Data Center (*Nucleic Acids Research*, 2020), Beijing Institute of Genomics (BIG), Chinese Academy of Sciences at https://bigd.big.ac.cn/gsa-human/HRA000339 under the approval of the Ministry of Science and Technology (accession number HDAC000233). Upon on reasonable request, the corresponding authors will provide additional help for accessing the sequence data. Data supporting the findings of this study are available within the article and its Supplementary information files and from the corresponding author upon reasonable request. A Reporting summary for this article is available as a Supplementary information file. Source data are provided with this paper.

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

## Acknowledgements

We thank Beijing IDMO Co., Ltd for establishing the PDX models and the help on the in vivo drug sensitivity tests. We also thank Ms. Yun Wang, Ms. Yanyan Hou, and Mr. Haifa Guo for their help in collecting patient samples and Dr. Kunkun Sun and Dr. Fei Yang for pathologic consultant. This work is funded by the National Key Research and Development Program of China (No. 2016YFC0900200) and the National Natural Science Foundation of China (Nos. 81771931 and 31971325).

## Author contributions

P.L., J.W., and X.C. designed the study; Y.H., F.S., Y.L., K.L., Z.C., Y.Z., and X.W., performed the experiments; X.S., F.Y., X.C., and Q.L. resected and offered cancer tissues from lung cancer patients; X.S. and Q.L. collected the clinical data. C.L. and B.Z. analyzed the whole-genome sequencing data. X.C., P.L., Y.H., and X.S. wrote the manuscript with the contributions of all the authors.

## Competing interests

The authors declare no competing interests.
