## [Peer Review File · Nature Communications]

REVIEWER COMMENTS

Reviewer #1 (Remarks to the Author):

In this study, Hu et al. showed various applications of lung cancer patient-derived organoids (PDO) with a focus on rapid drug response assay platform that allows the prediction of patient's response within a week. There is a significant overlap with recently published lung cancer organoid studies (Kim et al, Nat Comm, 2019; Sachs et al, EMBO J, 2019). However, this reviewer feels that the current study has a significant advancement over the previous two studies, especially with the microwell-based rapid drug test platform. Hence, I recommend its publication in Nature Communications with minor revision.

The authors pointed out two very important obstacles: currently the PDO-based drug test requires at least a few weeks or months to deliver the results and the success rate for lung cancer PDO establishment is still low or heavily affected by normal cell contamination. To circumvent these problems, the authors tried a radical approach by using minimal organoid media and shorten the assay time window in a week of time, which makes a sophisticated long-term culture condition rather obsolete. This reminds us of the former drug test trials using floating cancer spheroids where most primary cancer cells were dying rapidly even without any drug challenge due to the lack of appropriate niche factors and extracellular matrix. Interestingly, the authors' sub-optimal culture condition allows a better condition than those for spheroids but worse enough to prevent normal cell outgrowth. Finding this sub-optimal culture condition must have been very crucial to optimize the overall drug screening time and efficiency.

I only raise just a few points here:

1. The 79% success rate on page 5 is for the trial of the drug test on the InSMAR-chip. This is not the rate for successful PDO establishment, which would require long-term capacity. It would be nice to point this out more clearly in the text not to cause any confusion to readers.
2. On page 5, when describing the number of organoids the author showed the number as 15 ± 24 and 3277 ± 4619 . What are these numbers? And how come it can be a negative value (e.g. $15 - 24 = -9$)?
3. On page 8, 'FACs' shall be changed to 'FACS' or 'flow cytometry'.
4. How many drugs can be tested at once on the InSMAR-chip?
5. In Sup Fig 10a, the blank white cells in the Heat map would mean 'not tested'. Please change them to black or gray to indicate conditions that are not tested.

Reviewer #2 (Remarks to the Author):

This manuscript by Hu and Sui et al. describes a new and efficient methods for generating and screening lung cancer derived organoids using a microchip technique. The manuscript describes an indepth analysis of a new method of generating lung cancer organoids from patient material and discusses a potential new method for drug screening and compares this to conventual methods and clinical results.

This manuscript is thorough in their experimental set-up and controls and verifies biological and clinical data. The extensive work presented here is innovative and creates a large potential for fast and efficient drug screening of patient-derived organoid systems. Their comparison to clinical data is striking and shows great promise for organoid-based clinical predictions. Therefore, this manuscript should be published at Nature Communications but not before some minor adaptations.

1. The authors claim a more efficient method of generating LCOs by mechanical instead of enzymatic dissociation. The method of enzymatic dissociation used in this manuscript is however not explained in the methods section.

A remaining question which can be answered by adding the methods section is the following:

During isolation using mechanical shearing, organoids are generated from filtered clumps of 40-100µm. This limits the number of single cells in the solution. If this filtering step is not included in the enzymatic digestion, more single cells could be plated in enzymatic digestion compared to mechanical shearing. This difference could alter the efficiency of organoid formation and explain the differences. If this filtering was performed the statement by the authors remains true.

2. The authors have generated LCOs from 103 samples. Yet only 20 samples were able to grown in limited medium that was depleted of niche factors. The authors do not comment on the limited outgrowth in the tumour medium. The authors could test for tumour cell percentage in lines that are able to grow in limited medium with lines that can not and correlate this.

3. In figure 3g, the authors claim no statistical difference between LCO on chip and off chip. Figure 3e, however, shows large differences in growth rate between the two culture conditions.

4. In line 157, the authors state that the LCO transcriptome is comparable to the tissue from which it originates. While this statement is valid, it would be better explained if the differences between tumour and organoids were briefly explained.

5. The x-axis description in the legend states percentage while the axis itself shows fraction. This should be changed as well as the maximum indicator should be moved to the end of the axis.

6. Figure 1g would be more descriptive if a barplot or dotplot was used per organoid line using the stdev of the 20 counted organoids as error bar.

7. Figure 2D shows the number of passages but does not include a time or passage indicator on its axis. An x-axis would make the figure more descriptive.
8. In line 100, the authors refer to Supplementary Fig. 1b while the graph is presented in Main Fig. 1b.

Reviewer #3 (Remarks to the Author):

In this manuscript by Hu et al, they describe a new method for generating lung cancer organoids by mechanical tissue processing and one day suspension culture followed by culture in a microwell array chip (InSMAR-chip). Using InSMAR-chip, they generated LCOs within a week with 79% of success rate. They show LCOs maintained histological features of original tumour tissues and similarity in copy number variation and mutations of specific genes by whole genome sequencing in a limited cases. Interestingly, they demonstrated that the InSMAR-chip method is likely to evolve into a clinical test that predicts drug response in lung cancer patients by comparing drug responses in patients and matching LCO. I believe the most valuable thing about this study is that it is possible to cultivate LCOs in a short time of a week, thereby increasing the applicability to the clinic. However, LCOs in InSMAR-chip are not demonstrated that they maintain tissue organization differently from transient primary cancer cell culture. Therefore, the manuscript seems better suited as a technical report for microwell lung cancer chip rather than a research article.

Major claims

1. I believe the most valuable aspect of this study is LCO generation using InSMAR-chip within one week for clinical use. However, LCOs in InSMAR-chip are not demonstrated that they maintain tissue organization differently from transient primary cancer cell culture. All histopathologic and genomic analysis were performed using LCOs of concurrent conventional matrigel culture. Drug responses of cancer cells may be different according to culture condition of 2D, 3D or organoid as demonstrated by Han et al (Nature 580, 137-141, 2020). Short term culture of lung cancer in InSMAR-chip is not completely proved to be different from transient primary cancer cell culture, and rather have tissue structures by a hierarchy of stem cells and differentiated cancer cells.
2. In general, cancer organoids constituted only cancer cells as reported elsewhere. As showed in Figure 1 h, tumor purity in LCOs in this study is low and suggest a mixture of cancer cells and stromal cells rather than pure cancer cell. In this occasion, how convince drug response using InSMAR-chip represent response of cancer cells?
3. The genetic analysis of the organoids is limited in only 7 models. This is considerably limited in scope than the in-depth analyses of organoids cultures using InSMAR-chip. Sequencing a larger

number of models cultivated in InSMAR-chip would help to better understand if the models truly reflect the patients' tumors for co-clinical trial.

4. The study design of co-clinical trial is not clearly presented.

Minor comments

1. Some LCOs and NLSs seems to be in cell apoptosis (Especially, Fig 2e, LC97-O and BSDC1-O in Supplementary Fig.2a, and Supplementary Fig.6a). The authors should provide a more comprehensive view showing optimal growth of LCOs in all conditions. This should be done for both tumour and normal organoids.

2. In figure 2a, quality of photo of H&E stains were suboptimal and make it difficult to determine the structure tissues and corresponding organoids. TTF-1 staining of LC 96-T dose not show tumour cells. In squamous cell carcinoma, distinct cell borders and keratinization described in lane 144-145. P40 and P63 IHC is a duplication.

3. Whole genome sequencing of 7 cases is not sufficient to prove conservation of genomic changes in LCOs. How select this 7 cases? As seen in figure 2c, genetic alterations in this 7 cases are quite different from other reports of lung cancer mutation such as TCGA data. LCOs using for drug sensitivity test of targeted drugs were completely different from the list of WGS. Additionally, methods of genetic sequencing to identify very rare EGFR mutation and ALK fusion were not described. In figure 2b, genome-wide copy number variation will be better than gene specific CNV.

4. LC96-T does not seem to be cancer. Mutation profile and RNA expression are not matched in LC96-T and O.

5. In main text lane 216-217, LC 124-O described as EGFR E709K mutation. In figure 4d, the organoids described as harbouring EGFR L858R mutation. Which one is correct?.

6. In all images provided, some of organoids have big sizes and some of them have small sizes, even this size variant is shown in the same well. Rather, the organoid sizes in long-term cultured experiment (supplementary Fig 6) are more constant than the organoid sizes in drug test (Fig1 C). Furthermore, the authors showed that the growth rates of each organoid are very diverse (Fig.1g). Therefore, the authors should estimate drug response to LCOs after considering organoid sizes not only the culture duration.

7. In figure 3e. the authors noted that continuous growth and viability of LCOs in conventional microplate and InSMAR-chip were similar. However, in image of Fig.3e, LCOs of conventional microplate were bigger than LCOs in InSMAR-chip suggesting different growth rate.

8. In all figures, organoid numbers should be displayed.

9. Methods of generation of PDXs and PDXs derived organoids were not presented.

10. During one week drug sensitivity tests as the co-clinical trial for targeted therapy and chemotherapy, the author recruited 21 patients. Seventeen patients biopsied from primary lung

tumour surgically as described in Supp. table 3 and main text line 241. What clinical conditions made targeted therapy or chemotherapy in the operable stage? Adjuvant setting? There is no stage information. Why genomic sequencing data were presented only 12 patients in supplemental table 2 without method description of genetic test? Some cancer patients such as LC95, 96 and 137 were treated with tyrosine kinase inhibitor although EGFR mutation is absent. Is it valid as co-clinical trial?

11. For passaging of LCOs, the authors take too long time for dissolving matrigel (2-3 hours). Comparing to digestion solutions such as trypsin, dispase or accutase, dissolving time is longer than 6-8 times. Is there any reason?

ASSOCIATE PROFESSOR PENG LIU
DEPARTMENT OF BIOMEDICAL ENGINEERING
TSINGHUA UNIVERSITY SCHOOL OF MEDICINE
HAIDIAN DISTRICT BEIJING 100084 CHINA

PHONE: (86) -10-62798732
FAX: (86) -10-62798732
EMAIL: pliu@tsinghua.edu.cn

Dear the reviewers,

Thank you very much for your comments on our *Nature Communications* manuscript (Manuscript ID: NCOMMS-20-19498) entitled “Patient-derived organoids analyzed on a superhydrophobic microwell array for predicting drug response of lung cancer patients within a week”. Our changes and responses to the reviewers’ comments are summarized below and marked in the revised manuscript that has been electronically uploaded.

Reviewer: 1

1. The 79% success rate on page 5 is for the trial of the drug test on the InSMAR-chip. This is not the rate for successful PDO establishment, which would require long-term capacity. It would be nice to point this out more clearly in the text not to cause any confusion to readers.

We appreciate the reviewer’s suggestion. Yes, the 79% success rate is the percentage of lung tumor samples which can produce enough number of tumor organoids for performing the on-chip drug sensitivity test, not the rate for establishing the LCO lines. To clarify this point, we changed the wording “concluding a success rate of 79%” on p.5 l.118 to “concluding a 79% success rate of sample processing...”.

2. On page 5, when describing the number of organoids, the author showed the number as 15+/-24 and 3277+/-4619. What are these numbers? And how come it can be a negative value (e.g. 15 - 24 = -9)?

The numbers of organoids are represented as average number \pm standard deviation, which is now explained following the numbers on p.5. Since the numbers of organoids generated from lung tumor samples can vary from 0 to more than 10000 and the numbers of normal lung spheroids (NLSs) vary from 0 to more than 100, it is normal that the standard deviations are bigger than the corresponding average numbers.

3. On page 8, ‘FACs’ shall be changed to ‘FACS’ or ‘flow cytometry’.

Now the “FACs” on p.9 l.229 was changed to “FACS”.

4. How many drugs can be tested at once on the InSMAR-chip?

The InSMAR-chip used in this study contains an array of 108 microwells. We test each drug at eight

different concentrations with 3 repeats. Therefore, four drugs can be tested together on a single InSMAR-chip. We added this information on p.20 l.550: “Each drug was tested at eight different concentrations with three repeats. As a result, four drugs can be tested together on a single InSMAR-chip containing 108 microwells.”

5. In Sup Fig 10a, the blank white cells in the Heat map would mean ‘not tested’. Please change them to black or gray to indicate conditions that are not tested.

Thanks for the suggestion. We have changed the white cells in Supplementary Fig. 10a to brown. Since we added a new Supplementary Fig. 8, so now it's Supplementary Fig. 11a

Reviewer: 2

1. The authors claim a more efficient method of generating LCOs by mechanical instead of enzymatic dissociation. The method of enzymatic dissociation used in this manuscript is however not explained in the methods section. A remaining question which can be answered by adding the methods section is the following: During isolation using mechanical shearing, organoids are generated from filtered clumps of 40-100µm. This limits the number of single cells in the solution. If this filtering step is not included in the enzymatic digestion, more single cells could be plated in enzymatic digestion compared to mechanical shearing. This difference could alter the efficiency of organoid formation and explain the differences. If this filtering was performed the statement by the authors remains true.

We agree with the reviewer that more single cells could be plated using the enzymatic digestion without the filtering step compared to mechanical shearing and more organoids could be obtained using the enzymatic digestion. However, it usually takes much longer time for single cells to grow into tumor organoids. Therefore, these single cells cannot be used for the one-week drug sensitivity test and the filtering step with the 40-µm strainer is necessary for both the enzymatic digestion and the mechanical shearing to exclude the interference of these single cells to the drug tests. This is why we didn't test the enzymatic digestion without filtering in the current study. To make it clearer, we added the following sentences in the methods session on p.16, l.414: “During the enzymatic digestion, the small pieces of tumor tissue were digested in 1% collagenase for 1 or 2 hours, followed by the collection of cell clusters between 40 and 100 µm”.

2. The authors have generated LCOs from 103 samples. Yet only 20 samples were able to grown in limited medium that was depleted of niche factors. The authors do not comment on the limited outgrowth in the tumour medium. The authors could test for tumour cell percentage in lines that are able to grow in limited medium with lines that can not and correlate this.

We apologize for our misleading description. Actually, we have successfully generated LCOs from more than 100 lung tumor samples and most of the LCOs can grow and maintain viability in this limited medium. Since most LCOs were used for developing the drug sensitivity assay, here we only closely traced 20 LCO lines to quantify their 7-day growth rates. To clarify this issue, we added the following sentences on p.5 l.129: “We closely traced the LCOs generated from 20 samples to quantify the 7-day growth rates in this limited medium. The experiment was performed on the microwell array, so that the organoids can be individually observed and the growth rates can be precisely calculated. We found that 19 of these samples showed active growth (7-day growth rate > 1), although their growth rates are very diverse (Fig. 1f, g).”.

3. In figure 3g, the authors claim no statistical difference between LCO on chip and off chip. Figure 3e, however, shows large differences in growth rate between the two culture conditions.

Thanks for the comments and we apologize for the misleading images shown in Fig. 3e. Although the LCOs cultured on the chip looked smaller than those in the multi-well plate, we actually calculated their n-day growth rates by dividing the areas of LCOs at day n by the areas of the same LCOs at day 1. The 7-day growth rates of the on-chip and the off-chip LCOs were 5.2 and 4.3, respectively, and the 14-day growth rates of the on-chip and the off-chip LCOs were 11.9 and 12.2, respectively. As illustrated in Fig. 3g, we quantified the growth rates of 4 LCO lines and found no statistical difference between LCOs cultured on the chip and in the multi-well plate. In order to avoid confusion to the readers, we changed the images of the LCOs cultured on the chip in Fig. 3e.

4. In line 157, the authors state that the LCO transcriptome is comparable to the tissue from which it originates. While this statement is valid, it would be better explained if the differences between tumour and organoids were briefly explained.

We performed the KEGG analysis following the reviewer's suggestion. The results were added in Supplementary Fig. 5b and the following description of the results was added on p.7 1.163: "*Kyoto Encyclopedia of Genes and Genomes (KEGG) analysis of the differential genes between LCOs and tumor tissues demonstrated enrichment of genes corresponding to cell adhesion molecules and the immune responses (Supplementary Fig 5b), consistent with the lack of a tumor microenvironment in organoid culture.*"

5. The x-axis description in the legend states percentage while the axis itself shows fraction. This should be changed as well as the maximum indicator should be moved to the end of the axis.

We appreciate the reviewer's comments. The word "percentage" in the legends of Fig. 1d, h was changed to "fraction" and the maximum indicator was moved to the end of the axis in Fig. 1d.

6. Figure 1g would be more descriptive if a barplot or dotplot was used per organoid line using the stdev of the 20 counted organoids as error bar.

Fig. 1g was changed to a barplot as the reviewer suggested.

7. Figure 2D shows the number of passages but does not include a time or passage indicator on its axis. An x-axis would make the figure more descriptive.

Now the passage indicator was added in Fig. 2d.

8. In line 100, the authors refer to Supplementary Fig. 1b while the graph is presented in Main Fig. 1b.

We are sorry for the mistake. The error was corrected.

Reviewer: 3

Major claims

1. I believe the most valuable aspect of this study is LCO generation using InSMAR-chip within one week for clinical use. However, LCOs in InSMAR-chip are not demonstrated that they

maintain tissue organization differently from transient primary cancer cell culture. All histopathologic and genomic analysis were performed using LCOs of concurrent conventional matrigel culture. Drug responses of cancer cells may be different according to culture condition of 2D, 3D or organoid as demonstrated by Han et al (Nature 580, 137-141, 2020). Short term culture of lung cancer in InSMAR-chip is not completely proved to be different from transient primary cancer cell culture, and rather have tissue structures by a hierarchy of stem cells and differentiated cancer cells.

Thanks for the reviewer's comments. First of all, as we mentioned on p.8 1.203 "*In addition, since the volume of the microwells is over 400 nL, we found that matrigel containing a limited number of organoids can be easily loaded into each microwell to form a uniform droplet array using an electronic pipette operated in the multi-dispense mode (Fig. 3c). After that, up to 2.4 μ l of culture medium can be overlaid onto each matrigel droplet using the spot-cover method to assist the growth of organoids in this droplet culture mode.*", please allow us to point out that the LCOs cultured in the microwells of the InSMAR-chip were actually embedded in the matrigel, which are then overlaid with the lung cancer organoid culture medium (LCOM) just like those cultured in the conventional multi-well plates. We believe both the methods should be able to provide the LCOs with the exact same culture conditions, which are distinct from the transient primary cancer cell culture. In order to make it clearer, the sentences on p.8 1.206 were modified to the following "*After gelation of the matrigel solution, up to 2.4 μ l of culture medium can be overlaid onto each gel droplet using the spot-cover method to assist the growth of organoids in this droplet culture mode (Fig. 3c).*" and a cartoon image was included in Fig.3c to illustrate this culture method.

Second, as shown in Fig. 3, we compared the growth rates and the viability of 4 LCO lines (LC96, LC97, LC120, and LC141) cultured on the InSMAR-chip with those in the plates. We demonstrated that the LCOs cultured on the chip and in the plate have the same 7-day growth rates, the same morphologies, and the same viability. Actually, we have traced the growth of LCOs on the InSMAR-chip for 21 days, which showed continuous growth without decline. The results were shown in Supplementary Fig. 8. Additionally, we harvested the LCOs cultured on the chip for 7 days and performed the histology analysis. The H&E staining clearly showed the on-chip LCOs were similar to the off-chip cultures, both of which maintained the structure of the original tumor tissue (Fig. 3i). Meanwhile, we found that when the LCOs were seeded on the chip or in the plate without matrigel, cells soon attached to the bottom to form a monolayer, losing the organized 3D structures. These results clearly demonstrated that our on-chip cultured LCOs were different from the transient culture of primary cancer cells. The following sentences were added p.8 1.207 "*LCOs cultured on the chip showed similar growth rates and viability as that cultured in the conventional microplate (Fig. 3e-h), and more importantly, maintained the 3D structures of the parental tumor tissue (Fig. 3i). We traced the growth of LCOs on the InSMAR-chip and observed continuous growth for more than 3 weeks without decline (Supplementary Fig. 8).*"

2. In general, cancer organoids constituted only cancer cells as reported elsewhere. As showed in Figure 1 h, tumor purity in LCOs in this study is low and suggest a mixture of cancer cells and stromal cells rather than pure cancer cell. In this occasion, how convince drug response using InSMAR-chip represent response of cancer cells?

We appreciate the reviewer's comment. First of all, we would like to mention that many previous studies have demonstrated the purities of cancer cells in tumor organoids could vary in a wide range. For example, Kopper *et al.* showed that the purity of cancer cells in ovarian cancer organoids was in the range of $88\% \pm 23\%$ (Kopper *et al.*, Nat. Med. 25:838-849, 2019), similar to that in our study ($78 \pm 17\%$). Vlachogiannis *et al.* demonstrated the purity of organoids derived from gastrointestinal cancers

varied from 60% to 95% (Vlachogiannis *et al.*, Science 359:920-926, 2018). Therefore, we believe there is not a clear definition on how pure the cancer organoids should be.

Second, stromal cells in the *in vivo* tumors can interact with cancer cells and thus affect patients' responses to anti-cancer therapies. A recent study demonstrated that stromal cells played an important role in the assembly of patient tumor clusters (PTCs) which recapitulate patients' responses in high accuracy, while cell clusters containing only the tumor epithelial cells showed distinct drug response patterns (Yin *et al.*, Sci. Transl. Med. 12:eaaz1723, 2020). Therefore, it is still unclear whether organoid cultures of pure cancer cells are more accurate than those mixed with stromal cells in predicting patients' responses to anti-cancer therapies or not.

Third, it was pointed out by some reports that overgrowth of normal organoids or stromal cells should be prevented in the culture of cancer organoids (Sachs and Clevers, Curr. Opin. Genet. Dev. 24:68-73, 2014), especially for lung cancer organoids (Dijkstra *et al.*, Cell Rep. 31:107588, 2020). Previous studies on lung cancer organoids employed the culture medium either with Netlin-3a to select lung tumor cells (Sachs *et al.*, EMBO J. 38:e100300, 2019), which was effective only in the TP53 mutant samples, or without the niche factors necessary for normal lung epithelial cells (Kim *et al.*, Nat. Commun. 10:3991, 2019). In the current study, we developed a mechanical sample processing method, which can significantly increase the purity of tumor cells in the P0 organoids as shown in Fig. 1h. In addition, we used a limited medium which does not contain the niche factors necessary for the growth of normal lung epithelial cells. As a result, the percentage of cancer cells in the organoids increased with passaging (Fig. 2f) and the overgrowth of normal organoids or stromal cells was not observed in our LCO cultures.

In addition, we demonstrated that LCOs cultured on the InSMAR-chip responded to the targeted drugs under the expected mechanisms (Fig. 4d,e) and in accordance to the genetic mutations of the original tumors (Fig. 5a,b), indicating that the responses of tumor organoids to targeted drugs did come from the cancer cells as the normal cells do not have the drug sensitive mutant and will not respond to the targeted drug. We also demonstrated that the results of the on-chip drug sensitivity tests were in good agreement with the PDX model (Fig. 4 i,j, Fig. 6f, Supplementary Fig. 10) and the clinical data (Fig. 5 c,d, Fig. 6 b,c, Supplementary Fig. 11d), further demonstrating the accuracy of the on-chip organoid-based drug sensitivity assay.

References:

- Kopper, O., *et al.* An organoid platform for ovarian cancer captures intra- and interpatient heterogeneity. Nat. Med. 25: 838-849 (2019).
- Vlachogiannis, G., *et al.* Patient-derived organoids model treatment response of metastatic gastrointestinal cancers. Science 359: 920-926 (2018).
- Yin, S., *et al.* Patient-derived tumor-like cell clusters for drug testing in cancer therapy. Sci. Transl. Med. 12: eaaz1723 (2020).
- Sachs, N., and Clevers, H. Organoid cultures for the analysis of cancer phenotypes. Curr. Opin. Genet. Dev. 24:68-73 (2014).
- Dijkstra, K., *et al.* Challenges in establishing pure lung cancer organoids limit their utility for personalized medicine. Cell Rep. 31: 107588 (2020).
- Sachs, N., *et al.* Long-term expanding human airway organoids for disease modeling. EMBO J. 38: e100300 (2019).
- Kim, M., *et al.* Patient-derived lung cancer organoids as in vitro cancer models for therapeutic screening. Nat. Commun. 10:3991 (2019).

3. The genetic analysis of the organoids is limited in only 7 models. This is considerably limited in scope than the in-depth analyses of organoids cultures using InSMAR-chip. Sequencing a larger number of models cultivated in InSMAR-chip would help to better understand if the models truly reflect the patients' tumors for co-clinical trial.

We sequenced five more organoids cultured on the InSMAR-chip as well as their corresponding tumor tissues. Just like what we expected, the results which have been added in Fig. 2 and Supplementary Fig. 4 further proved that the genetic alterations were consistent between the LCO and the tumor tissue pairs. The major goal of the current study is to develop a technically feasible means for predicting patient-specific drug responses using lung cancer organoids in a short turnaround time, not for establishing a biobank of LCOs. So we did not perform WGS for a large number of samples. Since we employed the P0 organoids, which were cultured less than a week in vitro, for the drug tests, we sequenced P0 organoids together with their corresponding tumor tissues, demonstrating all the pairs had the same patterns of the genetic alterations. Considering the very short culturing time (usually only a few days for the P0 organoids) and the extremely high cost for the WGS, we think these 12 examples should be sufficient to prove the models truly reflect the patients' tumors.

4. The study design of co-clinical trial is not clearly presented.

We appreciate the reviewer's question. We would like to apologize for the misuse of the words "co-clinical data" on p.2 l.36, which may cause the reviewer regarded our study as a co-clinical trial. Rather, the organoid-based drug sensitivity test presented in this manuscript is only a pilot study, not a co-clinical trial. We replaced the word "*co-clinical data*" with "*The results*" and the description of the study design was added in the Methods session on p.15: "*Study design. The objective of the study is to determine whether the on-chip LCO-based drug sensitivity test can represent the patients' responses to standard-of-care treatments. The collection of patient tissues and data for the generation of LCOs and the LCO-based drug sensitivity tests were approved by the ethical review boards of the Peking University People's Hospital. Main inclusion criteria included patients with clinically local advanced or metastatic lung cancer, aged 18 years or older, fresh tissues available through either biopsy or surgical resection of the primary or metastatic lesions, and enough numbers of organoids generated from patients' samples. Candidates were assessed to determine eligibility and informed consent were obtained before operation. The drug sensitivity tests were performed on the InSMAR-chips once the organoids were successfully established within a week after operation. However, only patients with concurrent or subsequent clinical drug response evaluations which were comparable with the on-chip assays were enrolled in the clinical study.*"

Minor comments

1. Some LCOs and NLSs seems to be in cell apoptosis (Especially, Fig 2e, LC97-O and BSDC1-O in Supplementary Fig.2a, and Supplementary Fig.6a). The authors should provide a more comprehensive view showing optimal growth of LCOs in all conditions. This should be done for both tumour and normal organoids.

Thanks for the reviewer's suggestion. LCOs showed a wide variation in morphologies and different levels of cell-cell adhesions. The LC97-O, a typical example of loosely connected granular sheets, maintained good viability and stably propagated for more than 20 passages. In order to show the optimal growth of the organoids, we replaced some of the images of LC97-O in Fig. 2e (P0 and P3), Supplementary Fig. 2a, and some of the images of the NLS in Supplementary Fig. 6a (P2 and P3). The image on the far right in Supplementary Fig. 2a was LC64-O not BSDC1-O, so we replaced it with the image of BSDC1-O. We apologize for the mistake.

2. In figure 2a, quality of photo of H&E stains were suboptimal and make it difficult to determine the structure tissues and corresponding organoids. TTF-1 staining of LC 96-T dose not show tumour cells. In squamous cell carcinoma, distinct cell borders and keratinization described in lane 144-145. P40 and P63 IHC is a duplication.

Thanks for the reviewer's comments to improve our manuscript. First, some of the H&E stains shown in Fig. 2a have been replaced with the images having higher magnificence to more clearly illustrate the structures of the tissues and the organoids.

Second, LC96 is a TTF-1 negative lung adenocarcinoma as indicated by the clinical data of the tumor marker stains, including CK7(+), TTF-1(-), NapsinA(-), p40(-), p63(-), CgA(-), Syn(-), CD56(-), Ki-67(60%+), CK20(-), CK(+). Both the LC96-O and the LC96-T were negatively stained with TTF-1, confirming the consistency between the LCOs and the parental tumor tissue in the gene expression of tumor markers.

Third, the squamous cell carcinoma LC97 has a low level of differentiation. Thus, the features of squamous cells, such as clear cell borders and keratinization, were not distinct in the H&E images of both the LC97-O and the LC97-T. In addition, following the reviewer's suggestion, we replaced the P63 stain of the LC97 with the CK5/6 stain. The description on p.6 l.145 was modified to: "*The squamous cell carcinoma organoids LC97-O showed low differentiation, high proliferation, positively expressed p40, and partial expression of CK5/6, recapitulating the features of the original tumor tissue*".

3. Whole genome sequencing of 7 cases is not sufficient to prove conservation of genomic changes in LCOs. How select this 7 cases? As seen in figure 2c, genetic alterations in this 7 cases are quite different from other reports of lung cancer mutation such as TCGA data. LCOs using for drug sensitivity test of targeted drugs were completely different from the list of WGS. Additionally, methods of genetic sequencing to identify very rare EGFR mutation and ALK fusion were not described. In figure 2b, genome-wide copy number variation will be better than gene specific CNV.

As we discussed in the above question, five more pairs of LCOs and tumor tissues were sequenced and analyzed, further proving the conservation of genomic changes in LCOs. We selected these samples (LC61-O, LC79-O, LC95-O, LC96-O, LC97-O, LC98-O, and LC99-O) for sequencing based on the quantities of P0 organoids, which should be enough for performing WGS, and the availability of normal tissues. Since these LCOs were not expanded *in vitro*, the sequencing experiments usually used up all the samples and no LCOs were left for the drug sensitivity tests. A new text was inserted on p.6 l.149: "*These samples were selected for sequencing based on the quantities of obtained organoids, which should be enough for performing WGS and RNA-seq, and the availability of normal tissues.*"

In Fig. 2c, we analyzed mutations in the top 30 most frequently mutated genes in lung cancer from the cBioPortal for Cancer Genomics (<http://cbioportal.org>) (Cerami *et al*, Cancer Discovery, 2012) which are not completely consistent with the most mutated genes in TCGA data since these two databases use different reference genome (TCGA uses hg38 while cBioPortal uses hg19) and different methods for sequence data processing. In addition, cBioPortal includes data on small cell lung cancer while TCGA only has non-small cell lung cancer sequences. Since we used hg19 as the reference genome and we included a small cell lung cancer sample, we used the cBioPortal gene list. To clarify this, we explained in the legend of Fig 2c "*Overview of somatic alterations in the top 30 most frequently mutated genes in lung cancer in the cBioPortal of Cancer Genomics (<http://cbioportal.org>)*".

After demonstrating that the LCOs recapitulate the genetic features of original tumors, new samples that have enough number of organoids were used to perform the drug sensitivity tests. WGS of these samples was not done because of the high costs. But the information of cancer gene mutations can be obtained from the clinical data of the patients who ordered genetic analysis. The clinical data of the genetic alterations including EGFR mutation and ALK fusion were identified using a capture-based targeted sequencing panel that consisted of 520 or 168 cancer-related genes (Burning Rock Biotech, Guangzhou, China), 457 or 31 cancer-related genes (Berry Oncology, Fujian, China) by next-generation sequencing as previously described (Chen *et al.*, Clin. Cancer Res. 25:7058-7067, 2019). A brief explanation was added on p.10 l.251: “*We first demonstrated that the responses of the LCOs to the targeted therapy were correlated to the genetic mutations of the original tumors using 12 samples³⁰, in which the information on genetic alterations including EGFR and ALK mutations can be obtained from the clinical data (Supplementary Table 2)*” and in the method session p.18 l.478 “*The clinical data of the genetic alterations including EGFR mutation and ALK fusion were identified using a capture-based targeted sequencing panel that consisted of 520 or 168 cancer-related genes (Burning Rock Biotech, Guangzhou, China), 457 or 31 cancer-related genes (Berry Oncology, Fujian, China) by next-generation sequencing as previously described⁵¹.*”

Finally, the gene specific CNV in Fig. 2b was replaced with the genome-wide copy number variation. Thanks!

References:

Cerami *et al.* The cBio Cancer Genomics Portal: An Open Platform for Exploring Multidimensional Cancer Genomics Data. Cancer Discovery. May 2012 2; 401.
Chen, K., *et al.* Perioperative dynamic changes in circulating tumor DNA in patients with lung cancer. Clin. Cancer Res. 25:7058-7067 (2019)

4. LC96-T does not seem to be cancer. Mutation profile and RNA expression are not matched in LC96-T and O.

We think the reason that caused the differences of the mutation profiles and RNA expressions between the LC96-T and the LC96-O is the low cancer cell purity in the original tumor tissue (45%) and the significantly increased purity in the generated organoids (88%). Therefore, distinct signal amplifications can be seen in the genome-wide CNVs and mutational signatures. Similarly, the correlation in gene expressions between the LC96-O and the LC96-T is also low (Fig. 2, Supplementary Fig. 4, and Supplementary Fig. 5). The same trends can be observed in the LC98 and the LC79, where the cancer cell purities in organoids were also markedly higher than those in parental tissues. By contrast, as the cancer cell purities in organoids of the LC97 and the LC61 were close to those in tumor tissues, the mutational profiles and the gene expressions in the LCOs and the tumor tissues are matched very well. A brief explanation was added on p.6 l.157: “*In the samples where cancer cell purity increased drastically in the LCOs compared to the original tissue (LC96-O, LC98-O, and LC79-O), more differences can be seen in the mutational profile between the organoids and the parental tissues.*”

Although the difference exists, we believe the LC96-T is a cancer rather than a normal tissue due to the following reasons. First, the correlation heat map of gene expressions in Supplementary Fig. 5a suggests that the LC96-T is clustered with tumor tissues rather than normal tissues and related more closely to the LC96-O than to the LC96-N (normal tissue from the same patient). Second, the copy number variations and the gene mutations were detected in LC96-T (Fig. 2c and Supplementary Fig 4b, c). The H&E and IHC stains of the LC96-T show the presence of cancer cells, although the fraction is low.

5. In main text line 216-217, LC 124-O described as EGFR E709K mutation. In figure 4d, the organoids described as harbouring EGFR L858R mutation. Which one is correct?

After double-checking the clinical data, we realized that the LC124-O has the EGFR P.G719A mutation. The description on p.9 1.227 was changed to “...LC124-O which harbors an EGFR P.G719A mutation at Exon 18...” and the legend of Fig. 4d were revised. Thanks a lot for the reviewer’s reminder and we are very sorry for the mistakes.

6. In all images provided, some of organoids have big sizes and some of them have small sizes, even this size variant is shown in the same well. Rather, the organoid sizes in long-term cultured experiment (supplementary Fig 6) are more constant than the organoid sizes in drug test (Fig1 C). Furthermore, the authors showed that the growth rates of each organoid are very diverse (Fig.1g). Therefore, the authors should estimate drug response to LCOs after considering organoid sizes not only the culture duration.

We agree with the reviewer that the sizes of the P0 organoids are quite heterogeneous, which may cause variations in the drug sensitivity tests. Therefore, in order to eliminate the variations, we adopted the following two measures: i) the viability of the LCOs was measured both before and after the drug treatment, the ratio of the two measurements was used to evaluate the drug effect. ii) eight different concentrations with three repeats were tested for each drug. As shown in Fig. 5 and 6, the variations among the repeats are acceptable and in good agreement with genetic mutations, PDX models, and clinical data. To further clarify this point, the text on p.8 1.213 was modified to: “*To eliminate the variations caused by the uneven numbers and the sizes of the P0 organoids seeded in the microwells, the viability of the LCOs was measured both before (AB-1) and after (AB-2) the drug treatment. The relative cell viability is then represented by the ratio of AB-2 over AB-1 and employed to evaluate the drug responses of LCOs.*”

We agree that the long-term culture may eliminate the variations due to the multiple times of passaging. However, as reported by previous studies, the long turnaround time of the assay and the risk of losing cancer cells by the normal cells will hurdle the application of LCOs in precision treatment of lung cancer (Ooft *et al.*, *Sci. Transl. Med.* 11, 2574, 2019; Dijkstra *et al.*, *Cell Rep.* 31, 107588, 2020).

References:

Ooft, S., *et al.* Patient-derived organoids can predict response to chemotherapy in metastatic colorectal cancer patients. *Sci. Transl. Med.* 11, 2574 (2019).

Dijkstra, K., *et al.* Challenges in establishing pure lung cancer organoids limit their utility for personalized medicine. *Cell Rep.* 31, 107588 (2020).

7. In figure 3e. the authors noted that continuous growth and viability of LCOs in conventional microplate and InSMAR-chip were similar. However, in image of Fig.3e, LCOs of conventional microplate were bigger than LCOs in InSMAR-chip suggesting different growth rate.

Please refer to our answer to the question 3 asked by reviewer 2.

8. In all figures, organoid numbers should be displayed.

We added the organoid numbers in Fig. 3 and Fig. 4.

9. Methods of generation of PDXs and PDXs derived organoids were not presented.

Methods of generation of PDXs are described in the Methods session on p.21 1.571-593. The methods of generation PDX derived organoids are described on p.22 1.595-600.

10. During one week drug sensitivity tests as the co-clinical trial for targeted therapy and chemotherapy, the author recruited 21 patients. Seventeen patients biopsied from primary lung tumour surgically as described in Supp. table 3 and main text lane 241. What clinical conditions made targeted therapy or chemotherapy in the operable stage? Adjuvant setting? There is no stage information. Why genomic sequencing data were presented only 12 patents in supplemental table 2 without method description of genetic test? Some cancer patients such as LC95, 96 and 137 were treated with tyrosine kinase inhibitor although EGFR mutation is absent. Is it valid as co-clinical trial?

We appreciate the reviewer's comment. The staging information has been listed in Supplementary Table 1. Among all the patients who received primary lung tumor removal, eleven patients with resectable stage I-III disease underwent radical resection, six patients with incurable stage IV disease underwent palliative primary tumor removal. We tended to select treatment-naive patients with unresectable stage IV disease or local advanced disease in higher risk of recurrence. We collected the clinical drug response data and compared with the on-chip results. However, there is the timing issue that restricted the recruitment of patients. First, only the fresh tumor tissues could be used for the on-chip drug tests. Second, the tested drug regimens must be prospectively suggested by the clinicians within one week. Therefore, the patients were assessed and enrolled based on the clinical stage rather than the delayed pathologic stage, leading to the enrollment of some patients with early stage lung cancer in the study.

The detailed information is shown as follows: LC96, 100, 124-128, and 135-137 were derived from patients with early stage lung cancer. Adjuvant treatment were recommended for certain patients with stage II/III lung cancer after operations. LC124, 125, 128, and 136 received adjuvant chemotherapy. LC100 received post-operative radiotherapy and refused chemotherapy. LC126 with EGFR L858R mutation received adjuvant TKI treatment following the regimen introduced in the ADJUVANT/CTONG1104 study. LC96 and 127 suffered rapid relapse before administration of adjuvant therapy. LC97 was derived from a patient with local advanced lung cancer who received both neoadjuvant and adjuvant chemotherapy combined with surgical resection. In the group of the patients who received palliative primary tumor removal, LC101, 129, and 133 were derived from patients with ipsilateral pleural dissemination. Our previous study showed surgery could be beneficial for improved survival in this highly selected group (Li *et al.*, *Eur. J. Cardiothorac. Surg.*, 55:1121-1129, 2019). LC95 and 131 were derived from lung cancer patients with oligometastase. LC94 was also derived from lung cancer patients with oligometastase but was excluded as paraffin section showed combined small cell lung cancer.

Genomic sequencing of the tumor tissue was important for identification of the efficiency of targeted therapies. The genomic testing was recommended by the clinicians and decided by the patients as clinical routine practiced. In total, 12 out of the 21 patients decided to take the genomic testing and the results were listed in Supplementary Table 2. The tissue DNA were profiled using a capture-based targeted sequencing that consisted a panel of 520 or 168 cancer-related genes (Burning Rock Biotech, Guangzhou, China), 457 or 31 cancer-related genes (Berry Oncology, Fujian, China) by next generation sequencing as previously described (Chen *et al.*, *Clin. Cancer Res.* 25:7058-7067, 2019).

We didn't know the genomic information of LC95, 96, and 137 when we performed the drug tests. All

the drug regimens tested in the study was pre-planned. The genomic testing usually come out several days later. Normally more drug regimens were tested on LCOs than taken by the patient. Therefore, our study is an observational study rather than a co-clinical trial. We apologize again for misuse this word.

References:

Li, H., *et al.* Primary tumour resection in non-small-cell lung cancer patients with ipsilateral pleural dissemination (M1a): a population-based study. *Eur. J. Cardiothorac. Surg.* 55:1121-1129 (2019).

Zhong, W., *et al.* Gefitinib versus vinorelbine plus cisplatin as adjuvant treatment for stage II-III A (N1-N2) EGFR-mutant NSCLC (ADJUVANT/CTONG1104): a randomised, open-label, phase 3 study. *Lancet Oncol.* 19:139-148 (2018).

Chen, K., *et al.* Perioperative dynamic changes in circulating tumor DNA in patients with lung cancer. *Clin. Cancer Res.* 25:7058-7067 (2019).

11. For passaging of LCOs, the authors take too long time for dissolving matrigel (2-3 hours). Comparing to digestion solutions such as trypsin, dispase or accutase, dissolving time is longer than 6-8 times. Is there any reason?

As we described on p.16 1.434 “*Briefly, LCOs were harvested and digested in 10× volumes of cold Organoid Harvesting Solution (R&D Systems) on an orbital shaker at 0 °C for 2-3 h to dissolving the matrigel.*”, we passaged the LCOs using the Organoid Harvesting Solution (R&D Systems), which is a non-enzymatic method for dissolving matrigel according to the manufacturer’s instruction. Intact organoids can be harvested for passaging while the matrigel can be digested completely. Actually, the digestion time can be shortened to 30-45 minutes if the gel is small. We have tried trypsin and collagenase in the preliminary study, but found it difficult to achieve the best results using the same digestion time. In some cases, the LCOs were digested to single cells within a few minutes while the matrigel was not degraded completely. Therefore, we chose the Organoid Harvesting Solution.

After this revision, we hope that you will find our revised manuscript suitable for publication in *Nature Communications*.

Sincerely yours,

Peng Liu
Associate Professor of Biomedical Engineering

REVIEWER COMMENTS

Reviewer #1 (Remarks to the Author):

I do not have any further comments and wish that the paper can be published in Nat Comm.

Reviewer #2 (Remarks to the Author):

Dear editor and authors,

The additional data added to the Nature Communications manuscript (Manuscript ID: NCOMMS-20-19498) entitled "Patient-derived organoids analyzed on a superhydrophobic microwell array for predicting drug response of lung cancer patients within a week" has answered all my questions. Therefore I would recommend the manuscript as currently uploaded ready for publication.

Below I would like to quickly comment on each point separately:

1. Since the authors state that the size filtering was performed on both mechanical as enzymatic digestion, the comparison made in outgrowth efficiency of LCOs is valid. This was achieved by adding the required method section. While seeding single cells could improve organoid number over time, the authors use the technique to quickly generate organoids which is not achieved by plating single cells.
2. After clarification in the text, the authors explain that the generation efficiency was higher than understood from the first text. By adding some extra lines on the generation of the organoids and the selection of a subset of organoids for further analysis, the process indeed is efficient.
3. By changing the representative images in 3e, the graph and images are now comparable.
4. The differences between the LCOs and tumor tissue is now explained in text and as figure. The question raised about these differences is therefore answered.
5. X-axis title is changed correctly
6. Change of plot 1g has clarified the data.
7. Addition of x-axis is correct

8. Textual reference mistake was corrected

Reviewer #3 (Remarks to the Author):

1. Vast majority of your 144 lung cancer sample listed in Suppl table 1 is adenocarcinoma histology. It is well known that most frequently mutated gene in Asian adenocarcinoma is EGFR and followed by TP53. KRAS mutation is also known as a major oncogenic alteration in adenocarcinoma. Therefore many lab uses a reference gene for quality control of genomic laboratory. Your data in figure 2, however, do not show EGFR and KRAS mutation. More over, major oncogenic pathway genes were not listed except TP53. In this occasion, how can you convince your bioinformatic analysis is correct?

2. I do not agree with the authors' opinion. In contrast to pluripotent stem cell-derived organoids that model development, adult stem cell-derived epithelial organoids recapitulate adult tissue repair (Clevers H Cell 165, 2016). Consequently, ASC-derived organoids can be established only from tissue compartments with regenerative capacity. At present, essentially all ASC-derived organoid types represent only the epithelial parts of organs, and there is an absence of stroma, nerves, and vasculature. (Schutgens F and Clevers H, Ann Rev Pathol 2020, 15:211-234). Lung carcinoma is a kind of epithelial neoplasm having regenerative capacity most likely from cancer stem cells. Therefore lung cancer stem cells also could not produce stroma, nerves, and vasculature during organoid generation. The purity of cancer cell study in your study ($78\pm 17\%$) may suggests stromal cell contamination during organoid culture rather than components of organoid. In this case, fibroblasts usually coexist in the metrigel admixed with organoids and ultimately inhibit organoid growth. Your photos of organoids also do not show any stromal components. If you convince stromal components or normal epithelial cells in the tumor organoids, you should show stromal components more straightforward way such as photos with IHC instead of bioinformatic estimation as Kopper et al and Valchogiannis et al did. In the case of stromal contamination is exist in each microwell of InSMAR-chip, how can you control the proportion of cancer/stromal cell proportion in each microwell to consistently measure drug response with high accuracy ?

3. All other concerns are addressed properly.

ASSOCIATE PROFESSOR PENG LIU
DEPARTMENT OF BIOMEDICAL ENGINEERING
TSINGHUA UNIVERSITY SCHOOL OF MEDICINE
HAIDIAN DISTRICT BEIJING 100084 CHINA

PHONE: (86) -10-62798732
FAX: (86) -10-62798732
EMAIL: pliu@tsinghua.edu.cn

Dear the reviewer,

Thank you very much for your comments on our revised *Nature Communications* manuscript (Manuscript ID: NCOMMS-20-19498A) entitled “Patient-derived organoids analyzed on a superhydrophobic microwell array for predicting drug response of lung cancer patients within a week”. Our changes and responses to your comments are summarized below and marked in the revised manuscript that has been electronically uploaded.

1. Vast majority of your 144 lung cancer sample listed in Suppl table 1 is adenocarcinoma histology. It is well known that most frequently mutated gene in Asian adenocarcinoma is EGFR and followed by TP53. KRAS mutation is also known as a major oncogenic alteration in adenocarcinoma. Therefore many lab uses a reference gene for quality control of genomic laboratory. Your data in figure 2, however, do not show EGFR and KRAS mutation. Moreover, major oncogenic pathway genes were not listed except TP53. In this occasion, how can you convince your bioinformatic analysis is correct?

Thanks for the comments to improve our manuscript. We have made changes in Fig. 2c, which now shows the mutations of frequently mutated lung cancer genes and oncogenic pathway genes, including EGFR and KRAS.

2. I do not agree with the authors’ opinion. In contrast to pluripotent stem cell-derived organoids that model development, adult stem cell-derived epithelial organoids recapitulate adult tissue repair (Clevers H Cell 165, 2016). Consequently, ASC-derived organoids can be established only from tissue compartments with regenerative capacity. At present, essentially all ASC-derived organoid types represent only the epithelial parts of organs, and there is an absence of stroma, nerves, and vasculature. (Schutgens F and Clevers H, Ann Rev Pathol 2020, 15:211-234). Lung carcinoma is a kind of epithelial neoplasm having regenerative capacity most likely from cancer stem cells. Therefore lung cancer stem cells also could not produce stroma, nerves, and vasculature during organoid generation. The purity of cancer cell study in your study ($78 \pm 17\%$) may suggests stromal cell contamination during organoid culture rather than components of organoid. In this case, fibroblasts usually coexist in the metrigel admixed with organoids and ultimately inhibit organoid growth. Your photos of organoids also do not show any stromal components. If you convince stromal components or normal epithelial cells in the tumor organoids, you should show stromal components more straightforward way such as photos with IHC instead of bioinformatic estimation as Kopper et al and Valchogiannis et al did. In the case of stromal contamination is exist in each microwell of InSMAR-chip, how can you control the proportion of cancer/stromal cell proportion in each

microwell to consistently measure drug response with high accuracy?

We appreciate the detailed and enlightening explanation provided by the reviewer. After carefully reading the papers mentioned in the comments, we totally agree with the reviewer that the lung cancer organoids in our study are epithelial and do not have the stromal components. We did observe the contamination of stromal cells in the surrounding matrigel in some cases as shown in the following images (the white arrows indicate the fibroblasts). In our study, we collected the organoids by filtering through 40 and 100 μm filters (i.e. only the cell clusters in the range of 40-100 μm were collected), which can largely decrease the numbers of stromal cells since they generally do not aggregate with other cells. We found this filtration step is not sufficient to remove all the stromal cells, but can make sure that the majority of cells in the microwells are the tumor organoids. In addition, we used a limited medium which does not support the growth of normal lung organoids to inhibit the growth of normal epithelial cells. As a result, our organoid cultures were cancer cell dominant although not 100% pure.

To further decrease the proportion of stromal cells in the microwell, we are now developing an image-based automatic organoid picking system which can pick individual organoids and seed them in the microwells directly without getting stromal cells. In addition, a quality control step of morphology checking should be performed to get rid of the microwells or samples with severe stromal contaminations in the future applications. To reflect these points, we added the following sentences in the discussion session in p.13 l.340: *“In order to consistently measure drug responses with a high accuracy, the contamination of normal cells in the LCO culture, including both the normal lung epithelial cells and stromal cells such as fibroblasts, should be avoided. In the current study, we collected organoids by filtering the samples through 40- and 100- μm filters (i.e. only the cell clusters in the range of 40-100 μm were collected) which can largely decrease the numbers of stromal cells since they generally do not aggregate with other cells. We used a limited medium which inhibits the growths of normal lung organoids and normal epithelial cells. As a result, our organoid cultures were cancer cell dominant although not 100% pure. In the future, an image-based automatic organoid picking system which can transfer individual organoids into the microwells directly without getting stromal cells can be used to further increase the purity of the LCOs in the InSMAR-chip. In addition, a quality control step of morphology checking should be performed to get rid of the microwells or samples with severe stromal cell contamination.”*

Figure: Images of lung tumor organoids with stromal cells, which are indicated by the white arrows.

After this revision, we hope that you will find our revised manuscript suitable for publication in *Nature Communications*.

Sincerely yours,

Peng Liu

Peng Liu

Associate Professor of Biomedical Engineering